# Ferromagnetic Fe-TiO$_2$ spin catalysts for enhanced ammonia electrosynthesis

Jingnan Wang[1,2,8], Kaiheng Zhao[3,8], Yongbin Yao[4], Fan Xue[4], Fei Lu[5], Wensheng Yan [6], Fangli Yuan[7] & Xi Wang [4] ✉

Magnetic field effects (MFE) of ferromagnetic spin electrocatalysts have attracted significant attention due to their potential to enhance catalytic activity under an external magnetic field. However, no ferromagnetic spin catalysts have demonstrated MFE in the electrocatalytic reduction of nitrate for ammonia (NO$_3$RR), a pioneering approach towards NH$_3$ production involving the conversion from diamagnetic NO$^{3-}$ to paramagnetic NO. Here, we report the ferromagnetic Fe-TiO$_2$ to investigate MFE on NO$_3$RR. Fe-TiO$_2$ possesses a high density of atomically dispersed Fe sites and exhibits an intermediate-spin state, resulting in magnetic ordering through ferromagnetism. Assisted by a magnetic field, Fe-TiO$_2$ achieves a Faradaic efficiency (FE) of up to 97% and an NH$_3$ yield of 24.69 mg mg$_{cat}^{-1}$ at −0.5 V versus reversible hydrogen electrode. Compared to conditions without an external magnetic field, the FE and NH$_3$ yield for Fe-TiO$_2$ under an external magnetic field is increased by ∼21.8% and ∼ 3.1 times, respectively. In-situ characterization and theoretical calculations show that spin polarization enhances the critical step of NO hydrogenation to NOH by optimizing electron transfer pathways between Fe and NO, significantly boosting NO$_3$RR activity.

The effectiveness of spin state regulation in enhancing the catalytic activities of active sites on catalysts has been demonstrated in specific reactions[1–4]. Significant progress has been made in the development of novel catalysts, such as heterogeneous geminal-atom catalysts[5], La-promoted Co nanoparticles[6], and Zn-based metal-organic framework (MOF) incorporating Co species[7]. Taking the electrochemical reduction of nitrate (NO$_3$RR)[8–15] as an example, remarkable advancements have been achieved in improving electrochemical performance through spin state control due to the magnetic transition occurring during the reduction process from diamagnetic NO$^{3-}$ to paramagnetic NO[16]. Zhang's group[17] achieved spin regulation of Fe-Ti pairs by manipulating oxygen vacancies on a monolithic Ti electrode, resulting

in high NH$_3$ yield (272,000 µg h$^{-1}$ mg$_{Fe}^{-1}$ at −0.4 V vs. RHE) and high Faradaic efficiency (95.2% at −0.4 V vs. RHE). Subsequently, Zhang, Xu, Seh, and coworkers[18] demonstrated that Li incorporation could regulate a spin-dependent Cu-Co high-entropy oxide, highlighting the crucial role played by high-spin-state cobalt in ammonia synthesis and emphasizing the broad impact of spin effects in electrocatalysis.

Recently, there has been a significant research focus on the magnetic field effects (MFE) in certain electrocatalytic reactions such as the oxygen evolution reaction (OER) and carbon dioxide reduction reaction (CO$_2$RR), particularly for ferromagnetic spin catalysts (FS-CTs). The interaction between FS-CTs and an external magnetic field influences the electron spin distribution within the catalysts, thus

[1]Institute of Molecular Engineering Plus, College of Chemistry, Fuzhou University, Fuzhou 350108, China. [2]State Key Laboratory of Heavy Oil Processing, China University of Petroleum, Beijing 102249, China. [3]Key Laboratory of Photochemistry, Institute of Chemistry, Chinese Academy of Sciences, Beijing 100190, China. [4]Key Laboratory of Luminescence and Optical Information, Ministry of Education, School of Physical Science and Engineering, Beijing Jiaotong University, Beijing 100044, China. [5]College of Physical Science and Technology, Yangzhou University, Yangzhou 225002, China. [6]National Synchrotron Radiation Laboratory, University of Science and Technology of China, Hefei, Anhui 230026, China. [7]State Key Laboratory of Mesoscience and Engineering, Institute of Process Engineering, Chinese Academy of Sciences (CAS), Beijing 100190, China. [8]These authors contributed equally: Jingnan Wang, Kaiheng Zhao. ✉e-mail: xiwang@bjtu.edu.cn

exerting a substantial impact on the catalytic process[19]. This approach enables precise regulation of the spin states of catalytic centers without introducing complexity to the system, presenting a straightforward and effective strategy for enhancing electrocatalytic performance. For instance, Lu's group[20] synthesized a series of room-temperature ferromagnetic Ni/MoS$_2$ single-atom catalysts (SACs) and observed that an external magnetic field significantly enhanced their OER performance. Sun's group[6] optimized the structure and spin configuration of ferromagnetic NiFe-LDHs through Cu$^{2+}$ doping, demonstrating that magnetic field assistance yields ultra-low overpotential, thereby enhancing OER performance. Sheng and Zhang et al. [21]. demonstrated that Cu* sites in oxide-derived copper catalysts exhibit magnetism and that an external magnetic field reduces the activation energy for CO$_2$RR C-C coupling, thereby enhancing the selectivity for C$_2$ products. However, to date, there have been no reported ferromagnetic spin catalysts demonstrating MFE in NO$_3$RR.

Among the NO$_3$RR catalysts[22–27], ferromagnetic Fe-based spin catalysts are regarded as promising models for investigating the magnetic field effects (MFE) due to their abundant iron reserves and diverse properties; however, their synthesis presents big challenges. In this study, we successfully synthesized ferromagnetic Fe-TiO$_2$ spin catalysts comprising high-density single-atom Fe species as model catalysts to explore the MFE in NO$_3$RR. Magnetization curves (M-H), physical property measurement system (PPMS), and X-ray emission spectroscopy (XES) analyses confirmed that Fe-TiO$_2$ was ferromagnetic and the Fe species within it exhibited an intermediate spin state. Under an external magnetic field, both Faradaic efficiency (FE) and NH$_3$ yield of Fe-TiO$_2$ were significantly enhanced: at −0.5 V vs RHE, FE increased from 80% to 97%, while NH$_3$ yield surged from 5.95 mg mg$_{cat}$$^{-1}$ h$^{-1}$ to 24.69 mg mg$_{cat}$$^{-1}$ h$^{-1}$. Operando Fe K-edge x-ray absorption near edge structure (XANES) spectroscopy, in-situ attenuated total reflectance fourier transform infrared spectroscopy (ATR-FTIR), and theoretical calculations revealed that the application of a magnetic field optimized the electronic structures between Fe active sites and

*NO intermediates by facilitating *NO reduction processes. This work provides valuable insights into understanding MFE observed in ferromagnetic spin catalysts during NO$_3$RR.

## Results

### Synthesis and characterization of Fe-TiO$_2$ catalysts

Based on our previously published work[28], we employed a modified method to prepare ferromagnetic Fe-TiO$_2$ spin catalysts. Supplementary Fig. 1 illustrates the synthesis of bulk TiO$_2$ nanosheets with atomically dispersed Fe through an ion-exchange process. Subsequently, we utilized a soft template method involving tetrabutylammonium hydroxide (TBAOH) to exfoliate the material, resulting in ultrathin-layered TiO$_2$ nanosheets doped with single-atom Fe (Fe-TiO$_2$). To accurately determine the Fe loading, analysis was conducted using an inductively coupled plasma optical emission spectrometer (ICP-OES), which revealed a final actual loading of 6.80 wt%.

The Fe-TiO$_2$ catalysts were initially characterized using X-ray diffraction (XRD), as shown in Supplementary Fig. 2. The preservation of the TiO$_2$ layered structure was confirmed by the presence of characteristic diffraction peaks at (010), (020), and (030)[29]. Notably, a slight shift towards higher diffraction angles in the (010) peak after Fe doping was observed, which can be attributed to the smaller ionic radius of Fe compared to Ti. This results in a contraction of the TiO$_2$ lattice according to Bragg's Law. To further elucidate the catalyst's features, scanning electron microscopy (SEM) images (Supplementary Fig. 3) and high-resolution transmission electron microscopy (HRTEM) images (Supplementary Fig. 4) were employed, revealing the ultrathin nanosheet morphology of the catalyst. The successful integration of Fe into the TiO$_2$ monolayer via ion exchange was confirmed by analyzing the energy-dispersive X-ray spectroscopy (EDS) spectrum shown in Fig. 1a, demonstrating its uniform dispersion throughout the sample. Aberration-corrected high-angle annular dark-field scanning transmission electron microscopy (AC-STEM) imaging technique combined

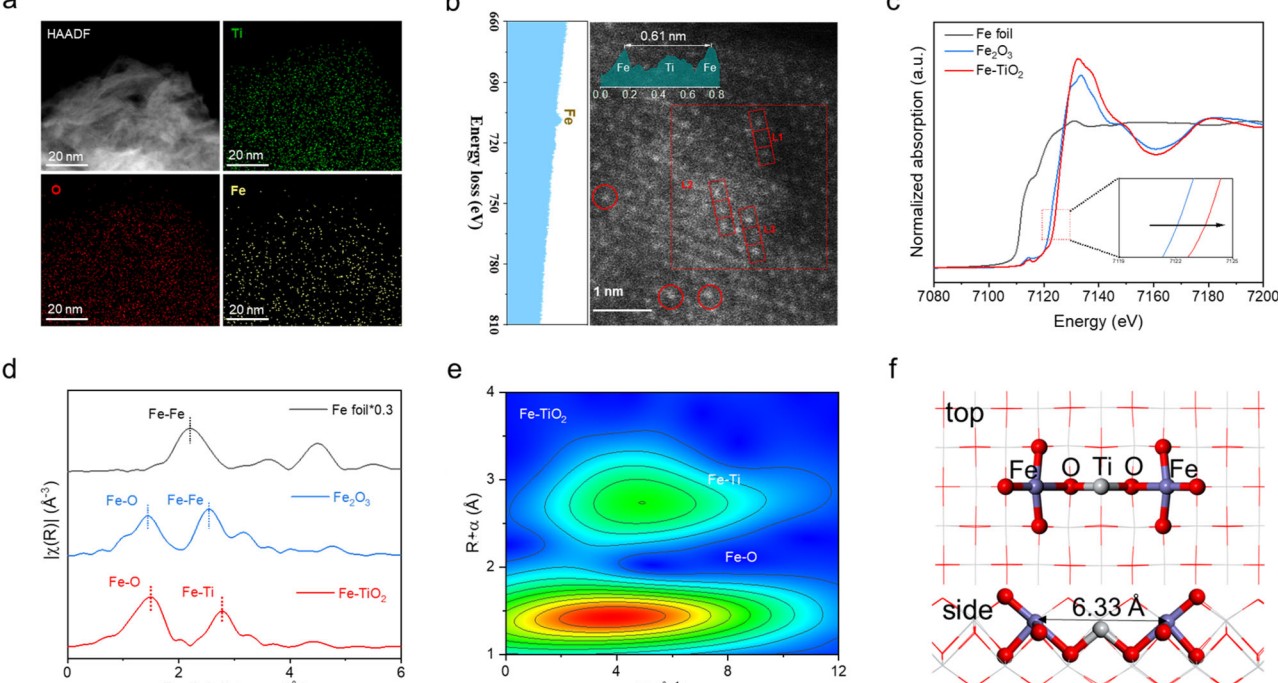

**Fig. 1 | Structural characterization of Fe-TiO$_2$ catalysts. a** EDS elemental mapping of Fe-TiO$_2$ catalysts. **b** Aberration-corrected HAADF-STEM image, where a single Fe atom is circled in red color. The inset on the left displays the energy loss spectrum of the selected dashed square. The inset above shows the intensity profiles. **c** Normalized Fe K-edge XANES spectra of Fe foil, Fe$_2$O$_3$ and Fe-TiO$_2$. The inset shows a local magnification. **d** Fourier transform of EXAFS spectra of Fe foil, Fe$_2$O$_3$ and Fe-TiO$_2$. **e** Wavelet transform EXAFS of the $k^2$-weighted $k$ space of Fe-TiO$_2$ catalyst. **f** The side and top views of the Fe-TiO$_2$ model. Source.

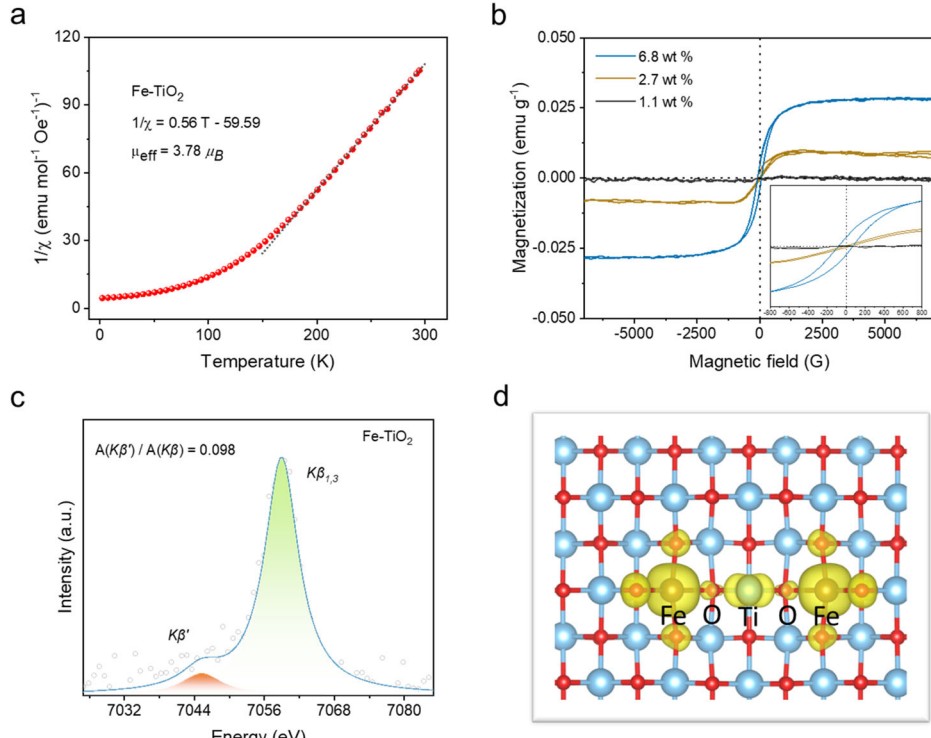

**Fig. 2 | Ferromagnetism of Fe-TiO$_2$ catalysts. a** Temperature-dependent inverse susceptibility $1/\chi$ for Fe-TiO$_2$ catalysts at 1000 Oe. **b** Magnetization curves (M-H) of Fe-TiO$_2$ with different Fe contents at 300 K. Insets show the coercivity of Fe-TiO$_2$ with different Fe contents at 300 K. **c** The Fe XES and corresponding peak area fitting of Fe-TiO$_2$ (6.80 wt%). **d** The spin density diagram of Fe-TiO$_2$ with spin alignment catalyst. The isosurface value is set to be 0.005 e/Bohr$^3$. Source data are provided as a Source Data file.

with electron energy loss spectrum analysis further revealed that Fe atoms have been seamlessly substituted for Ti within the atomic-level dispersed TiO$_2$ lattice in Fe-TiO$_2$ (Fig. 1b). The oxidation state assessment of Fe-TiO$_2$ was conducted through normalized XANES spectrum analysis (Fig. 1c). It is worth noting that there is a slightly lower absorption threshold at Fe K-edge in Fe-TiO$_2$ compared to that of Fe$_2$O$_3$, indicating a slightly higher oxidation state (+3 or above) of iron ions incorporated into TiO$_2$. This result aligns with our previous XPS (Supplementary Fig. 5). Furthermore, the XPS analysis results also indicate that Fe in Fe-TiO$_2$ is in an unsaturated coordination state (Supplementary Fig. 6).

The local coordination environment of Fe was further investigated, and detailed insights into the finer structure of the Fe-O coordination were obtained via Fourier-transformed (FT) extended XAFS spectroscopy (Fig. 1d, Supplementary Fig. 8, and Supplementary Table 1). The EXAFS spectrum revealed a prominent scattering peak at 1.53 Å, corresponding to the first shell of the Fe−O bond, with a fitted Fe-O coordination number of 5.0. Importantly, no scattering peaks indicative of Fe−Fe bonds were detected at 2.21 Å within the Fe-TiO$_2$. These observations collectively indicate that Fe atoms within Fe-TiO$_2$ are atomically dispersed with a coordination number of five with oxygen atoms (FeO$_5$). Furthermore, the presence of a Fe-Ti bond at 2.79 Å in the second shell suggests that Ti is substituted by Fe and incorporated into the TiO$_2$ lattice structure, forming a −Ti−O−Fe−O−Ti− coordination environment configuration involving both Fe-O and Fe -Ti interactions as evidenced by wavelet transformed EXAFS data results (Fig. 1e). Integrating these analytical results with our synthesis methodology suggests the Fe-O structure adopts an imperfect octahedral geometry characterized by the presence of oxygen vacancies (Fig. 1f).

## Magnetic property of Fe-TiO$_2$

As research progresses, an increasing body of evidence suggests a strong correlation between the electronic structure and magnetic properties of catalyst active sites and their catalytic performance[30–32]. Therefore, we performed a comprehensive analysis of Fe-TiO$_2$ catalysts. Temperature-dependent magnetic susceptibility curves were obtained for Fe-TiO$_2$ and Fe$_2$O$_3$ under a constant external magnetic field of 1000 Oe (Supplementary Figs. 9 and 10a). Figure 2a shows the inversion susceptibility $(1/\chi)$ plot of Fe-TiO$_2$. By utilizing the Curie-Weiss law[33,34], we calculated the Curie constant (C) and Weiss constant (θ) from which an effective magnetic moment ($\mu_{eff}$) of 3.78 $\mu_B$ for Fe-TiO$_2$ was determined, lower than that of Fe$_2$O$_3$ at 5.46 $\mu_B$ (Supplementary Fig. 10b). Considering that $\mu_{eff}$ is associated with the number of unpaired electrons per formula unit and high-spin Fe$_2$O$_3$ possesses five unpaired electrons, it can be suggested that fewer than five unpaired electrons in iron species are present in catalysts such as Fe-TiO$_2$. Utilizing the formula $\mu_{eff} = g*(J(J+1))^{1/2}$ (where g represents the Landé factor, typically equal to two, and J denotes the total angular momentum)[35], reveals that for three unpaired electrons, the corresponding value of $\mu_{eff}$ is approximately 3.87 $\mu_B$, closely matching our observed value of 3.78 $\mu_B$ for Fe-TiO$_2$, suggesting its possession of three unpaired electrons.

Subsequently, the magnetization curves of the Fe-TiO$_2$ catalysts were measured using a vibrating sample magnetometer (Supplementary Fig. 11a). In contrast to the typical diamagnetic behavior observed in TiO$_2$ (Supplementary Fig. 12), the incorporation of Fe into TiO$_2$ resulted in a combination of ferromagnetic and paramagnetic behavior within Fe-TiO$_2$. To gain deeper insights into the ferromagnetic properties exhibited by Fe-TiO$_2$, catalysts with different Fe loadings were synthesized, and their magnetization curves were fitted using the Langevin mode[36], yielding characteristic ferromagnetic magnetization curves for different loadings of Fe-TiO$_2$ catalysts (Fig. 2b). It is evident that as the loading increases, Fe-TiO$_2$ gradually exhibits ferromagnetic characteristics. This phenomenon can be attributed to the coupling of localized magnetic moments associated with high doping densities of Fe atoms, leading to significant contributions from $3d$ electrons

toward these magnetic moments[37]. Notably, 6.80 wt% Fe-TiO$_2$ is highly responsive to magnetic fields (subsequent mentions of Fe-TiO$_2$ refer to 6.80 wt% Fe-TiO$_2$). To further elucidate spin states, X-ray emission spectroscopy (XES) is commonly employed for investigating material spin states. Previous studies have indicated a linear relationship between the relative area under $k\beta'$ peak compared to that under the entire $k\beta$ peak and the compound spin state[38]. Accordingly, we collected XES data for several materials possessing similar covalent bonds (FePC, FeO, Fe(acac)$_3$) as shown in Supplementary Fig. 13. By fitting the peak areas corresponding to $k\beta'$ (approximately at 7044 eV) and $k\beta_{1,3}$ (approximately at 7058 eV), we generated a standard curve representing spin states (Supplementary Fig. 14). The XES data for Fe-TiO$_2$ (Fig. 2c) revealed the presence of a $k\beta_{1,3}$ satellite peak, designated as $k\beta'$, originating from the $3p$-$3d$ orbital interactions around 7045 eV, indicating an average Fe spin state above zero. By fitting the relative area of $k\beta'$, we determined its spin state to be 1.6, corresponding to three unpaired electrons. Therefore, based on these characterizations mentioned above, we have successfully demonstrated that the spin state of Fe in Fe-TiO$_2$ can be classified as an intermediate spin state (Supplementary Fig. 15). According to literature reports[39], the interaction between Fe and the TiO$_2$ carrier, as well as the reduction in crystal field splitting energy caused by unsaturation in Fe's coordination, leads to a transition of Fe's spin state from the conventional high-spin state (as observed in Fe$_2$O$_3$) to an intermediate-spin state. Specifically, only one electron occupies an orbital from eg orbitals. This implies that during the adsorption process of intermediates, Fe sites neither excessively bind to intermediates due to unoccupied $e_g$ orbitals nor fail to adsorb intermediates due to fully occupied $e_g$ orbitals.

Concurrently, we employed density functional theory (DFT) calculations to gain atomic-level insights into the origin of magnetism in Fe-TiO$_2$. Initially, our spin-polarized DFT calculations revealed the presence of unpaired Fe $3d$ electrons on the Fe-TiO$_2$, inducing spin polarization on neighboring O and Ti atoms. Consequently, an overall net spin value of approximately 3.523 $\mu_B$ was observed surrounding the Fe-O-Ti species (Supplementary Figs. 16 and 17). The magnetic ground state of Fe-O-Ti originates from spin exchange interactions of flowing electrons within this region, as described by Stoner's criterion (Supplementary Figs. 18 and 19, projected density of states (PDOS) of corresponding Ti and O atoms).

With an increase in the Fe loading, we further constructed a configuration with two adjacent FeO$_5$ sites (Fig. 2d). The Fe-Fe distance in Fe-TiO$_2$ is 6.33 Å. Notably, the ferromagnetic coupling (Fe-TiO$_2$ with spin alignment) between these two Fe sites is favorable by 18.59 meV/atom over the antiferromagnetic coupling (Fe-TiO$_2$ without spin alignment) (Supplementary Fig. 20), consistent with the room temperature (298.15 K) ferromagnetism observed experimentally. The ferromagnetic ground state of Fe-TiO$_2$ is further reflected in the electronic structure for the Fe-O-Ti region, exhibiting asymmetry in the spin-up and spin-down electron densities near the Fermi level (Supplementary Figs. 21–23).

### Electrocatalytic NO$_3$RR performance and MFE over Fe-TiO$_2$

The performance of the Fe-TiO$_2$ catalysts for NO$_3$RR was determined using a three-electrode system in an H-type electrolytic cell, with 0.1 M KOH and 0.1 M KNO$_3$ as the electrolyte. Before performance testing, argon gas was purged for 30 minutes (min) to eliminate oxygen and nitrogen sources from the air. The reaction products were quantitatively analyzed using UV-visible and $^1$H NMR spectroscopy (Supplementary Figs. 24–26). The NO$_3$RR activity of the catalyst was determined through linear sweep voltammetry (LSV) (Supplementary Fig. 27a). Meanwhile, Fe-TiO$_2$ exhibited higher NH$_3$ yield rate and FE compared to TiO$_2$ (Supplementary Fig. 27b, c and Figs. 28–30). These findings suggest that Fe-TiO$_2$ facilitates more efficient NO$_3$RR.

Additionally, $^{15}$N isotope labeling experiments were conducted to determine the nitrogen (N) source in NH$_3$ (Supplementary Fig. 31).

These experiments demonstrated that NH$_3$ originated from the reduction of the added NO$_3^-$ rather than other contaminants (Supplementary Fig. 32). Precise quantification of NH$_4^+$ was achieved using UV-visible and $^1$H NMR techniques, with results showing significant similarity, thereby confirming the accuracy of both quantification methods (Supplementary Fig. 33). Electrochemical impedance spectroscopy (EIS) revealed that Fe-TiO$_2$ exhibited lower charge transfer resistance ($R_{ct}$) and faster charge transfer rates compared to TiO$_2$ (Supplementary Fig. 34)[40]. Cyclic voltammetry (CV) curves were employed to evaluate the double-layer capacitance ($C_{dl}$) of Fe-TiO$_2$ and TiO$_2$, providing insights into their electrochemically active surface area (ECSA)[14]. The findings demonstrated that the C$_{dl}$ of Fe-TiO$_2$ was nearly three times that of TiO$_2$, indicating a higher abundance of active sites (Supplementary Figs. 35, 36).

To investigate the presence and impact of an MFE during the NO$_3$RR, we subjected the working electrode surface to a constant perpendicular magnetic field strength of 300 mT based on prior experimental evidence (Fig. 3a and Supplementary Fig. 37). Initially, we performed a preliminary assessment of the magnetic field enhanced NO$_3$RR was conducted using LSV (Fig. 3a). Remarkably under an external magnetic field (Fe-TiO$_2$-M), the LSV curve shows significant changes compared to that without the external magnetic field (Fe-TiO$_2$). This application of an external magnetization led to significantly increased current densities indicative of heightened NO$_3$RR performance specifically attributed to its influence upon Fe-TiO$_2$ catalysts.

Subsequently, the chronoamperometry experiments were conducted at various potentials to evaluate the influence of the magnetic field on NH$_3$ yield rate and FE (Supplementary Fig. 38). Measurements of NH$_3$ yield rate and FE at different potentials confirmed an increase in both parameters with the application of the magnetic field (Fig. 3b, c and Supplementary Fig. 39). At −0.5 V vs. RHE, the application of an external magnetic field resulted in an enhancement of NH$_3$ yield rate from 5.95 mg mg$_{cat}^{-1}$ h$^{-1}$ to 24.69 mg mg$_{cat}^{-1}$ h$^{-1}$, while FE improved from 80 % to 97 %. At −0.6 V vs. RHE, an NH$_3$ yield rate of 30.15 mg mg$_{cat}^{-1}$ h$^{-1}$ was achieved, making Fe-TiO$_2$ one of the most promising catalysts reported to date (Fig. 3d). Additionally, we investigated the relationship between magnetic field strength and MFE, observing that as the magnetic field strength increased, both NH$_3$ yield rate and FE gradually improved (Supplementary Fig. 40). However, when the magnetic field strength exceeded 200 mT, both parameters ceased to change significantly due to reaching the saturation magnetization (Fig. 2b). Furthermore, even after significantly reducing nitrate concentration, substantial increases in NH$_3$ yield rate and FE were observed under external magnetic fields (Supplementary Fig. 41). Finally, a stability test consisting of one hundred cycles was performed at a potential of −0.5 V vs. RHE to evaluate Fe-TiO$_2$ catalyst's durability under the influence by a magnetic field during reaction conditions. As shown in Fig. 3e, the catalyst exhibited stability with no significant decline in catalytic activity after one hundred cycles. Considering the distinct magnetic properties associated with different Fe loading, we conducted separate tests to evaluate the NO$_3$RR performance corresponding to 1.1 wt% and 2.7 wt% Fe loading. Remarkably, our findings revealed a direct correlation between the catalyst's magnetic properties and its MFE. The minimal impact on NH$_3$ yield and Faradaic efficiency under an external magnetic field is exhibited by the 1.1 wt% Fe-TiO$_2$ catalyst, which shows only paramagnetism, significantly less than the effect observed for 6.8 wt% Fe-TiO$_2$ (Supplementary Figs. 42–44, Tables 3–5). This may be attributed to the optimization of mass transfer processes under the influence of Lorentz forces; however, the effect is considerably smaller than the significant enhancement observed in ferromagnetic materials (Supplementary Fig. 45, Tables 6, 7).

To investigate the origin of the MFE, CV curves were re-measured in a non-Faradaic region at various scan rates (Supplementary Fig. 46).

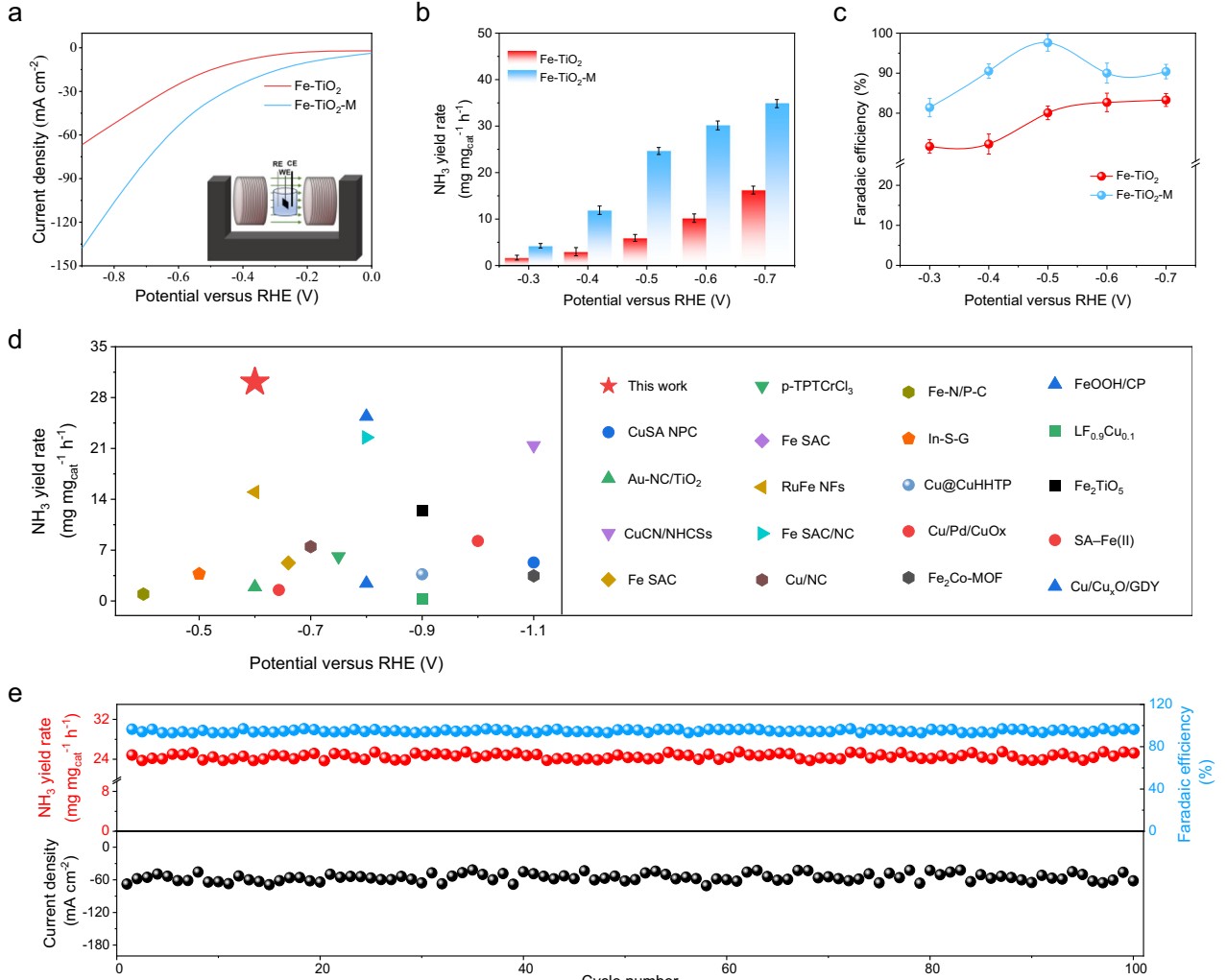

**Fig. 3 | MFE of ferromagnetic Fe-TiO₂ on electrocatalytic performance for NO₃RR. a** LSV curves were obtained for Fe-TiO₂ in the absence (red) and presence (blue) of an applied magnetic field. The inset shows a schematic diagram of the setup for electrocatalytic experiments under an applied magnetic field. Diagram of electrochemical testing under a magnetic field. NH₃ yield **b** and FE$_{NH3}$ **c** of Fe-TiO₂ catalysts at various potentials in the presence or absence of 300 mT external magnetic fields. **d** NH₃ yield comparison of Fe-TiO₂ with other reported electrocatalysts. The corresponding data are shown in Supporting Information Table 2. **e** The consecutive recycling stability tests at −0.5 V vs. RHE over the Fe-TiO₂ catalysts. All electrochemical data shown in the figures are not iR-corrected. Source data are provided as a Source Data file.

The observations indicated that the capacitance double layer (C$_{dl}$) of Fe-TiO₂ remained relatively stable, indicating the structural integrity of the catalyst under the applied magnetic field. This implies that intrinsic changes in the catalyst's structure did not contribute to the MFE. Subsequently, EIS measurements were conducted on Fe-TiO₂ under a magnetic field condition. The Nyquist plot (Supplementary Fig. 47) revealed a decrease in charge transfer resistance for Fe-TiO₂ under this condition, suggesting that the magnetic field facilitated and accelerated charge transfer during NO₃RR catalysis by Fe-TiO₂, thereby enhancing NH₃ production and FE.

Furthermore, for a comprehensive investigation of the magnetic effects in the NO₃RR process, it is crucial to consider not only the catalyst's magnetism but also the magnetic properties of the reactant species. The production of NH₃ involves sequential cleavage of N − O bonds and the formation of N − H bonds at the active site, which encompasses a pivotal intermediate, NO[11]. This transformation leads to a conversion from diamagnetic NO₃⁻ to paramagnetic NO. Analysis of the molecular orbitals reveals unpaired electrons within the antibonding π orbitals of NO, indicating its paramagnetic nature (Supplementary Fig. 48). Consequently, due to its susceptibility towards

externally applied magnetic fields, NO exhibits an essential role in investigating alongside MFE.

## Mechanistic insights into MFE of Fe-TiO₂

To investigate the impact of magnetic fields on Fe-TiO₂ catalysts and reaction intermediates, we conducted XANES spectroscopy measurements on their NO₃RR under both applied magnetic field conditions and without. Under a 100 mT magnetic field, the rising edge of the Fe K-edge in Fe-TiO₂ at working potential exhibits a more pronounced shift than that observed in the absence of a magnetic field (Fig. 4a, b). This observation suggests that within the catalysts, active sites comprising Fe species experience enhanced electron density accumulation under the influence of a magnetic field, thereby facilitating efficient electron transfer from Fe to reaction intermediates. Further analysis using XANES spectroscopy supports this characteristic by indicating stronger electronic hybridization between Fe-O-Ti when subjected to a magnetic field.

The synergy of magnetic fields is one of the primary methods reported for spin regulation[41–43]. To characterize the charge differences in Fe-TiO₂ with different spin configurations, the charge density

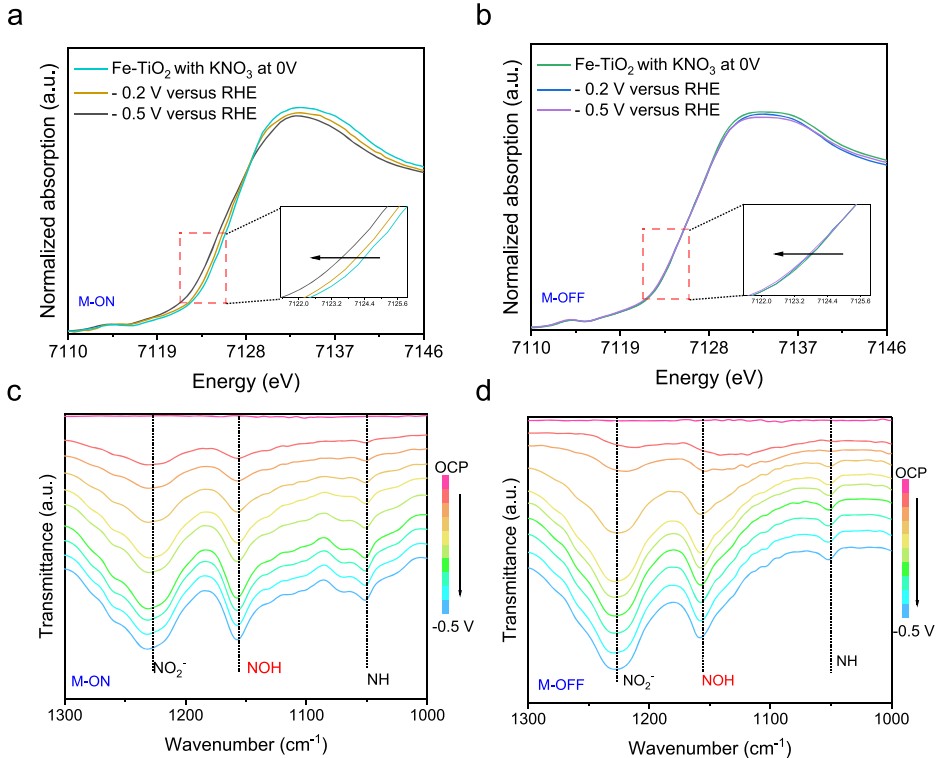

**Fig. 4 | Mechanistic studies for the NO₃RR over Fe-TiO₂.** Operando Fe K-edge XANES spectra for Fe-TiO₂ under 100 mT **a** and 0 mT **b**. The inset shows a local magnification. In-situ ATR-FTIR spectra of Fe-TiO₂ with negative scan from OCP to −0.5 V vs. RHE in the presence **c** or absence **d** of external magnetic fields. Source data are provided as a Source Data file.

differences of Fe-TiO₂ were calculated. It is indicated that the redistribution of d-orbital electrons in Fe atoms increases the electron density in certain orbitals, thereby enhancing their ability to transfer electrons to reaction intermediates (Supplementary Fig. 49). Additionally, the differential charge density of *NO and Bader charge analysis further demonstrate that the charge transfer between Fe and *NO can be regulated by different spin configurations (Supplementary Figs. 50 and 51).

To further investigate the influence of the magnetic field on the entire reaction process, in situ ATR-FTIR spectroscopy was employed to detect intermediates with and without an applied magnetic field. Initially, experiments were performed using TiO₂ and Fe-TiO₂ in 0.1 M KOH and 0.1 M KNO₃ electrolytes, covering a potential range from open circuit potential (OCP) to −0.5 V vs. RHE. For Fe-TiO₂ (Fig. 4d and Supplementary Fig. 52a), as the potential became more negative, a series of characteristic peaks corresponding to reaction intermediates appeared in the ATR-FTIR spectra: N − O stretching vibration of NO₂⁻ ions (1227 cm⁻¹)[15], NOH species (1156 cm⁻¹)[44], and NH species (1050 cm⁻¹)[45]. These peaks intensified progressively, indicating an enhancement in the reaction, consistent with electrochemical testing results. Concurrently, a series of weaker characteristic peaks were observed, including a negative peak at 1380 cm⁻¹(corresponding to the N − O stretching vibration of adsorbed NO₃⁻ ions)[46], the N − O bending vibration in NO at 1567 cm⁻¹[47], and the N − H bending vibration of NH₄⁺ ions at 1467 cm⁻¹ (Supplementary Fig. 53b)[46]. The gradual enhancement of these characteristic peaks indicates an intensifying reaction that is consistent with electrochemical testing observations and aligned with theoretical reaction steps. In contrast, for pure TiO₂ (Supplementary Fig. 54), weak characteristic peaks related to NO₃⁻ were observed through ATR-FTIR spectroscopy, implying inadequate adsorption capacity for pure TiO₂ towards NO₃⁻, thereby hindering completion of the reaction pathway (the presence of very weak characteristic peaks related to NH₄⁺ and other intermediates further

supports this). These results demonstrate that Fe serves as the exclusive active site for NO₃RR within Fe-TiO₂.

Subsequently, Fe-TiO₂ was further investigated to elucidate the underlying mechanism of the observed MFE (Fig. 4c and Supplementary Fig. 52b, 53a). Upon application of magnetic field, significant changes in the characteristic peaks of reaction intermediates were observed. The NO peak disappeared, the intensity of the *NOH species vibration at 1156 cm⁻¹ was notably increased, and the *NH species vibration at 1050 cm⁻¹ was also intensified. Additionally, a distinct *NH₂ peak appeared at 1439 cm⁻¹. These changes indicate that the reaction proceeded more rapidly under the influence of the magnetic field. It can thus be inferred that the magnetic field accelerated the hydrogenation process, converting *NO to *NOH, thereby expediting the overall reaction.

To elucidate the origin of the magnetic field effect (MFE) in Fe-TiO₂, various spin configurations of Fe-TiO₂, including spin states S = 0, 2, 4, 6, 8, and 10, were investigated[19,32,48]. These configurations, representing distinct magnetic states, were analyzed to facilitate an understanding of the role of MFE in the catalytic performance of Fe-TiO₂. Furthermore, the influence of spin configurations on material properties was systematically examined by calculating surface energies, reaction pathways, and energy changes. It was indicated that spin configurations significantly affect surface energy, with the high spin states (S = 8) configuration being the most stable (Supplementary Fig. 55). Additionally, the entire reaction pathways for different spin configurations were calculated. Significant energy differences for various intermediates were observed, with the high spin states (S = 8) configuration exhibiting the lowest reaction barrier, identifying it as the optimal spin configuration (Supplementary Fig. 56). Based on this, Gibbs free energy calculations for two representative magnetic structures, Fe-TiO₂ with spin alignment and Fe-TiO₂ without spin alignment, were performed, as shown in Fig. 5a[49]. Protonation of *NO (*NO + H⁺ + e⁻ → *NOH) is identified as the rate-determining step in Fe-TiO₂ without

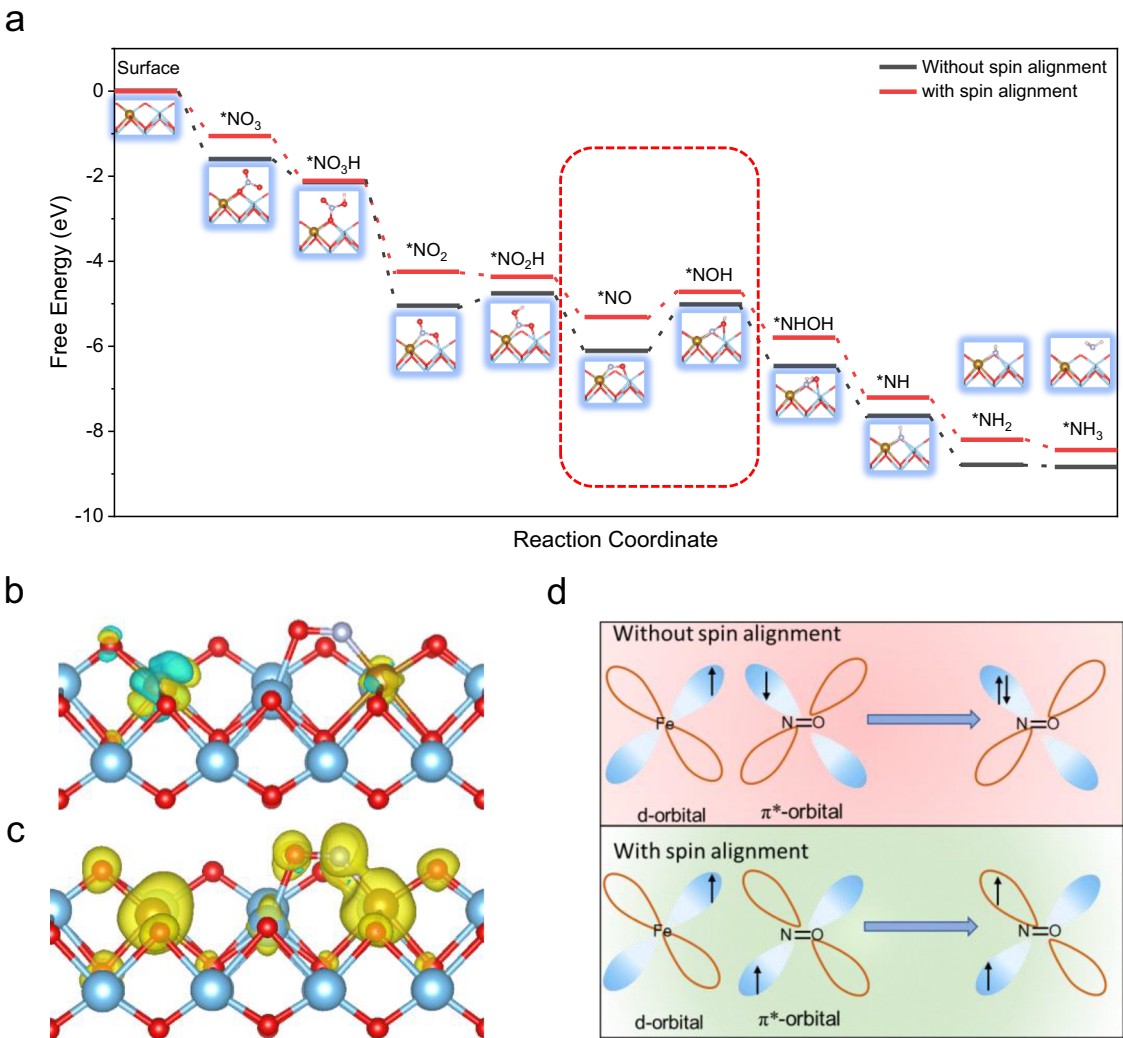

**Fig. 5 | DFT calculations results for the NO₃RR. a** Free energy diagram for nitrite electroreduction over Fe-TiO₂. The corresponding atomic coordinates are contained in Supplementary Data 1. The spin density of NO adsorbed-on **b** Fe-TiO₂ without spin alignment or **c** Fe-TiO₂ with spin alignment. The sky blue, red, and grayish white represent Ti, O and N atoms, respectively. **d** Schematic of spin-exchange mechanism for NO and Fe-TiO₂. Source data are provided as a Source Data file.

and with spin alignment cases. The energy for the rate-determining step at the Fe-TiO₂ with spin alignment active site is 0.59 eV, while the Fe-TiO₂ without spin alignment coupling presents a higher energy barrier of 1.09 eV. Subsequently, we calculated the charge density difference and the spin density for *NO. Compared with Fe-TiO₂ without spin alignment, Fe-TiO₂ with spin alignment exhibits stronger interaction strength between the active site and intermediates (Supplementary Fig. 51). Additionally, NO adsorbed-on Fe-TiO₂ without spin alignment shows a magnetic moment of 0μB compared to 0.7μB for Fe-TiO₂ with spin alignment, indicating distinct interaction mechanisms between Fe-TiO₂ with spin alignment and Fe-TiO₂ without spin alignment with NO (Fig. 5b, c). Based on these characteristics and theoretical analyses, it can inferred that the application of an external magnetic field induces spin polarization. Therefore, during the interaction between Fe and NO, electrons are inclined to be contributed to the unoccupied π* orbitals of NO. Since this process does not require overcoming the repulsion between electrons, it accelerates the activation of NO, thereby enhancing the overall catalytic activity (Fig. 5d). Therefore, the hydrogenation of NO molecules into NOH is more favorable at Fe-TiO₂ with spin alignment sites than at Fe-TiO₂ without spin alignment sites with an associated energy reduction by 0.50 eV resulting in a lower energy barrier for NO₃RR.

## Discussion

In summary, we have successfully synthesized a ferromagnetic Fe-TiO₂ spin catalyst composed of single-atom Fe species to investigate the MFE in NO₃RR. Various characterization techniques, including AC-HAADF-STEM and XAFS, have effectively confirmed Fe single-atoms within Fe-TiO₂. M-H, PPMS, and XES analysis have demonstrated that Fe-TiO₂ exhibits ferromagnetism with intermediate-spin state of Fe species. Importantly, compared to conditions without an external magnetic field, application of an external magnetic field resulted in ~21.8% increase in FE and 3.1-fold increase in NH₃ yield for Fe-TiO₂ catalysts. In-situ ATR-FTIR spectroscopy, Operando Fe K-edge XANES analysis, and electrochemical characterization revealed that the presence of a magnetic field expedites electron transfer processes and accelerates the NO reduction process. DFT calculations have demonstrated that spin polarization optimizes the electron transfer pathway between active sites and the crucial intermediate NO, which lays a solid foundation for its high activity towards NO₃RR.

## Methods

### Chemicals and materials

The reagents used in the synthesis and experimental processes were not further purified. Cesium carbonate (Cs₂CO₃), and titanium dioxide

($TiO_2$, Rutile) were purchased from Alfa Aesar. Ferric oxide ($\alpha$-$Fe_2O_3$) was from Acros. tetrabutylammonium hydroxide (TBAOH) and the ammonium ion standard solution (1000 µg/mL) were purchased from Aladdin. Potassium nitrate ($KNO_3$) and potassium hydroxide (KOH) were purchased from Shanghai Macklin.

## Catalyst synthesis

**Synthesis of $TiO_2$.** $TiO_2$ and $Cs_2CO_3$ were ground in a mortar at the mole ratio in the formula ($Cs_{0.7}Ti_{1.825}O_4$, CTO) for 30 min, followed by calcination at 800 °C for 20 h in air. Then the sample continued to be ground for 30 min and calcined at 800 °C for 20 h in the air to obtain layered titanate. The second step was protonation to form layered $TiO_2$ with extended interlayer distance. For every 1 g of titanate powder, it was dispersed in 100 mL of 1 M HCl solution. The HCl solution needed to be replaced three times every 24 h. Subsequently, $H_{0.7}Ti_{1.825}O_4$ was exfoliated by dispersing it in a 0.017 M TBAOH solution on a table concentrator for a duration of 10 days. The ratio of TBAOH solution to $H_{0.7}Ti_{1.825}O_4$ was maintained at 250 ml g$^{-1}$ throughout the process. Finally, the resulting residue was obtained through centrifugation at a speed of 20900 rpm min$^{-1}$ and thoroughly washed with ultrapure water until achieving neutral pH and removing excess TBAOH content. After freeze-drying for three days, fluffy $TiO_2$ ultra-thin nanosheets were successfully obtained.

## Synthesis of Fe-$TiO_2$

The synthesis process, as illustrated in Supplementary Fig. 1, closely resembled that of $TiO_2$ with the notable distinction lying in the precise combination of several raw materials ($TiO_2$, $Cs_2CO_3$, and $Fe_2O_3$) at a specific molar ratio. Notably, the metal component exhibited a molar fraction of 0.2.

## Material characterizations

The structure and morphology of several catalysts were characterized by X-ray diffraction (XRD, Bruker D2 diffractometer with Cu K$\alpha$ radiation), Scanning Electron Microscopy (SEM, Hitachi S-4800 field emission SEM), Transmission Electron Microscopic (TEM, Thermofisher Talos F200X with acceleration voltage of 200 kV), and High Angle Annular Dark Field Scanning Transmission Electron Microscopy (HAADF-STEM, FEI Tecnai G2 F20). The X-ray Photoelectron Spectroscopy (XPS) analysis was performed on a Thermo ESCALAB 250Xi electron spectrometer with 300 W Al KR radiation. The magnetic thermal curves were obtained at Physical Property Measurement System (PPMS-9, Quantum Design) to identify the spin state of the catalysts. The Fe $K\beta$ XES measurements were conducted at the 4W1B beamline of the Beijing Synchrotron Radiation Facility (BSRF). Fe K-edge X-ray absorption spectroscopy (XAS) was measured at the beamline 1W1B of the Beijing Synchrotron Radiation Facility (BSRF). Metal foils served as reference samples. The XAFS data analysis was conducted using Athena and Artemis within the Demeter software package. Initially, the XAFS raw data underwent energy calibration, background subtraction, and normalization. The XANES spectra were generated by subtracting the post-edge background from the overall absorption, followed by normalization relative to the edge jump step. Subsequently, $\chi$(k) data within the k-space range of 3.0 to 12.0 Å$^{-1}$ was Fourier-transformed into real (R) space employing Hanning windows (with a dk of 1.0 Å$^{-1}$) to distinguish the XANES contributions from various coordination shells. The amplitude reduction factor ($S_0^2 = 0.80$) was determined based on Fe foil and was consistently applied for the analysis of the other samples.

## In situ ATR-FTIR measurements

The catalyst ink was prepared by adding 0.005 g of catalyst and 30 µL of 5% Nafion solution into a mixed solution of 500 µL ethanol and 470 µL water, followed by ultrasonication for 1 h. A homemade in-situ infrared reaction cell was employed, utilizing a single-cell three-electrode system with a graphite rod as the counter electrode, a saturated

Ag/AgCl reference electrode, and the working electrode. The magnetic field was provided by a neodymium iron boron magnet, and the magnetic field strength applied to the catalyst surface was approximately 100 mT. Electrochemical testing was carried out using a CHI 660E electrochemical workstation (Chenhua), and all spectra were collected after stabilizing the corresponding potential for 30 min. The spectrometer used was a Thermo-Fisher Nicolet II spectrometer equipped with an MCT cryogenic detector.

During the collection of ATR-FTIR spectra, an attenuated total reflectance (ATR) prism made of wedge-shaped Si crystal was utilized, with an incident angle of 60 degrees. All measurements were obtained through 64 scans with a spectral resolution of 4 cm$^{-1}$. Chronoamperometry tests at OCP to −0.5 V RHE were conducted on a CHI 660E electrochemical workstation, and spectra were collected.

## Electrocatalytic nitrate reduction measurements

All electrochemical characterizations were carried out using the CHI 760E (Chenhua) electrochemical workstation. The test environment is air atmosphere at normal temperature and pressure. And All tests were not IR corrected. A three-electrode system was employed with a platinum foil serving as the counter electrode, a saturated Ag/AgCl reference electrode, and the working electrode. An H-type electrochemical cell was utilized, separated by a Nafion 117 proton-exchange membrane. Before use, the Nafion proton-exchange membrane was boiled in a 5% $H_2O_2$ aqueous solution at 80 °C for 1 h and subjected to multiple rinses. The carbon paper used was treated before usage with a mixed solution of $H_2SO_4$ and $H_2O_2$ (1:3, vol.) for 12 h, followed by several rinses to remove surface impurities. It was then trimmed to a size of 1*3 cm$^2$ for later use. The catalyst ink was prepared by adding 0.005 g of the catalyst and 30 µL of a 5% Nafion solution to a mixture of 500 µL ethanol and 470 µL water, followed by 1 h of sonication. The working electrode was created by evenly applying 20 µL of the catalyst ink (0.1×10$^{-3}$ g of catalyst) to carbon paper (0.3*1 cm$^2$), allowing it to air-dry naturally. The electrolyte solution was a mixture of 0.1 M KOH and 0.1 M $KNO_3$ which was stored at room temperature (298.15 K) and pressure. The potential was converted to RHE using the following equation: E (vs. RHE) = E (vs. Ag/AgCl) + 0.0591 × pH + 0.197 (pH = 13 ± 0.17 in 0.1 M KOH). Before testing, argon gas was introduced for 30 min to eliminate the influence of atmospheric nitrogen ($N_2$). Cyclic voltammetry (CV) was performed at a potential range of 0.3–0.5 V vs. RHE at a scan rate of 50 mV s$^{-1}$ to stabilize the electrode. Subsequently, linear sweep voltammetry (LSV) was carried out at a scan rate of 10 mV s$^{-1}$, followed by chronoamperometry (CA) testing.

For the stability cycling tests, the catalyst loading, coating area, electrolyte composition, and volume were kept identical to previous conditions. The test potential was set to 0.5 V, with an external magnetic field of 300 mT, using the same H-type electrolytic cell as before. Each test lasted for 1 h, after which the electrolyte was collected for colorimetric analysis and calculation of yield and Faradaic efficiency. Three samples were taken and analyzed, with the results averaged. This constituted one cycle. After each cycle, the electrode was gently rinsed with fresh electrolyte, and the next cycle was conducted with new electrolyte.

EIS measurements were conducted at open circuit potential with an applied AC voltage of 5 mV, over a frequency range of 0.5 Hz to 10 kHz. The electrolyte consisted of 0.1 M KOH and 0.1 M $KNO_3$.

## Determination of ammonia concentration

The generated $NH_3$ was quantitatively analyzed using the indophenol blue method. A certain volume of the post-reaction electrolyte was taken from the cathode chamber and diluted to be within the detection range. Then, 1 mL of the diluted electrolyte was mixed with 2 mL of

1.0 M NaOH solution (including 5 wt% salicylic acid and 5 wt% sodium citrate), followed by the addition of 1 mL of 0.05 M NaCl solution and 0.2 mL of 1 wt% sodium nitroprusside solution. After thorough mixing, the mixture was allowed to stand at room temperature (298.15 K) for 2 h. UV-Vis absorption spectroscopy data were collected at a wavelength of 655 nm. Subsequently, the concentration of $NH_3$ was calculated using a standard curve. All experiments were performed in triplicate.

For a more accurate quantification of $NH_3$ concentration, 1H NMR (600 MHz) was employed. A certain volume of the post-electrolysis solution was taken from the cathode chamber, and the pH was adjusted to 2 using a 0.05 M $H_2SO_4$ solution. Then, 0.9 mL of this solution was mixed with 0.1 mL of deuterated dimethyl sulfoxide (DMSO-D6), and after thorough mixing, it was subjected to NMR analysis. After integrating the characteristic peaks in the nuclear magnetic resonance spectrum, $NH_3$ concentration was calculated using a standard curve. All experiments were performed in triplicate.

### Determination of nitrite concentration

The detection of nitrite ions ($NO_2^-$) was performed using the Griess method. A mixture of p-aminobenzenesulfonamide (4 g), N-(1-naphthyl) ethylenediamine dihydrochloride (0.2 g), ultrapure water (50 mL), and phosphoric acid (10 mL, $\rho = 1.70$ g/mL) was used as the color reagent. A certain volume of the post-electrolysis solution was taken from the cathode chamber and diluted within the detection range. Then, 5 mL of the diluted electrolyte solution was mixed with 0.1 mL of the color reagent solution. After 30 min of incubation, UV-Vis absorption spectra were collected at a wavelength of 540 nm, and the concentration of $NO_2^-$ was subsequently calculated using a standard curve. All experiments were performed in triplicate.

### Calculation of the FE and $NH_3$ yield rate

The $NH_3$ yield rate ($\mu$g h$^{-1}$ mg$_{cat}^{-1}$) is calculated using the following formula:

$$Yield\ rate_{NH3} = (C_{NH3} \times V)/(t \times m_{cat}) \tag{1}$$

The Faradaic Efficiency (FE) for $NH_3$ is calculated using the following formula:

$$FE_{NH3} = \frac{8 \times C_{NH3} \times V \times F \times 10^{-6}}{17 \times Q} \times 100\% \tag{2}$$

The Faradaic Efficiency (FE) for $NO_2^-$ is calculated using the following formula:

$$FE_{NH3} = \frac{8 \times C_{NO2-} \times V \times F \times 10^{-6}}{46 \times Q} \times 100\% \tag{3}$$

In this equation, $C_{NH3}$ represents the concentration of $NH_3$ detected in the catholyte ($\mu$g mL$^{-1}$); $C_{NO2-}$ represents the concentration of $NO_2^-$ detected in the catholyte ($\mu$g mL$^{-1}$); V is the volume of the catholyte in the cathode compartment (30 mL); t stands for the reaction time (1 h); $m_{cat}$ indicates the mass of the loaded catalyst (mg); F is Faraday's constant (96485 C mol$^{-1}$); and Q refers to the total charge transferred during the reaction (C).

**$^{15}$N isotope labeling experiments.** The only difference from the above electrochemical testing procedure is the use of 99% K$^{15}NO_3$ as the nitrate source for isotope labeling experiments. These experiments were conducted at a potential of −0.6 V vs. RHE to determine the source of $NH_3$ in the products. The detection of products was carried out using the previously mentioned 1H NMR method.

### Computational details

All density functional theory (DFT) calculations were conducted using the Vienna Ab initio Simulation Package (VASP, version 5.4.4)[50,51]. The projected-augmented-wave (PAW) pseudopotential method was employed to describe the ion-electron interactions[52]. The electron exchange and correlation were described using the Perdew-Burke-Ernzerhof (PBE) functional within the generalized gradient approximation (GGA)[53,54]. The GGA + U calculations are performed using the Coulomb repulsion term U in the Hubbard Hamiltonian (DFT + U formalism) based on Dudarev's approach[55]. A kinetic energy cutoff of 500 eV was used for expanding wavefunctions over a plane-wave basis set. In all calculations, the electronic self-consistent calculations were performed using the 3 × 3 × 1 gamma-centered k-point mesh. During geometry optimizations, the threshold for the total energy and forces was set to 10$^{-5}$ eV and 0.02 eV Å$^{-1}$, respectively. A vacuum spacing of 20 Å was applied perpendicular to the $TiO_2$ surface to prevent periodic interactions. The equation gives the calculation of the free energy of adsorbed species: $G = E_{DFT} + E_{ZPE} - T\Delta S$, where $E_{DFT}$ is obtained from DFT energy, $E_{ZPE}$ and $T\Delta S$ of adsorbed species are computed through vibrational analysis, and thermodynamic corrections for gas molecules are based on standard databases.

## Data availability

The all data generated in this study are provided in the Supplementary Information. Supplementary Data 1 contains the atomic coordinates of the optimized computational model used in this work. Source data are provided with this paper.

## Code availability

All software used in this work is commercial software/open source software. No specific code was developed for this work.

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

## Acknowledgements

This work was supported by the Science and Technology Project of Guangdong Province (No. 2020B0101370001, X. W.). We thank the beamlines BL14W1 and BL11B at Shanghai Synchrotron Radiation Facility (SSRF), beamlines 1W1B at Beijing Synchrotron Radiation Facility (BSRF), and the Beamlines MCD-A and MCD-B (Soochow Beamline for Energy Materials) at National Synchrotron Radiation Laboratory (NSRL), and Anhui Absorption Spectroscopy Analysis Instrument Co, Ltd. for XAFS measurements and analysis. We thank the staff members at BL01B beamline of the National Facility for Protein Science in Shanghai (NFPS), Shanghai Advanced Research Institute, Chinese Academy of Sciences, for providing technical support and assistance in data collection and analysis. We also thank Dr. Baipeng Yin from the Institute of Chemistry, Chinese Academy of Sciences for his invaluable support and suggestions.

## Author contributions

X. W. conceived and designed the experiments. J.N. W., K.H. Z. and F. L. completed all characterizations. F. X., W.S. Y. and F.L. Y provided data for the analysis. J.N. W., K.H. Z. and Y.B. Y. carried out theoretical calculations, participated in the discussion and analysis, wrote the manuscript. All authors have approved the final version of the manuscript.

## Competing interests

The authors declare no competing interest.
