## [Transparent Peer Review file · Nature Communications]

Ferromagnetic Fe-TiO₂ spin catalysts for enhanced ammonia electrosynthesis

Corresponding Author: Professor Xi Wang

Version 0:

Reviewer comments:

Reviewer #1

(Remarks to the Author)

In this manuscript, the authors reported a new type of ferromagnetic spin electrocatalyst: Fe-embedded TiO₂ (Fe-TiO₂). They found that Fe-TiO₂ showed significantly higher NH₃ yield and Faradaic efficiency for the electrocatalytic reduction of nitrate (NO₃RR), under the influence of magnetic field effect (MFE). The authors have conducted comprehensive analysis for the Fe-TiO₂ structure and investigated the mechanism of MFE by the Fe-TiO₂. The research is a significant advancement on studying magnetic electrocatalysis for NO₃RR. I would suggest the manuscript to be accepted in the journal of Nature Communications, after a minor revision.

1. By XANFS, the authors found the pattern of -Ti-O-Fe-O-Ti- in the Fe-TiO₂ catalyst. Can other atomic configurations such as -Ti-O-Fe-O-Fe- and -Ti-O-Ti-O-Fe- be present as well? Please provide evidence for that. If so, how can you control the homogeneity of structure?
2. On Page 13, the authors mentioned "this observation suggests that within the catalysts, active sites comprising Fe species experience enhanced electron density accumulation under the influence of a magnetic field, thereby facilitating efficient electron transfer from Fe to reaction intermediates". Could the authors comment on why electron density accumulated on Fe under magnetic field? Can you link to computational evidence to support that?
3. Figure 5 shows the bonding configurations of reactant/product and intermediate species on Fe-TiO₂ surface. Do you have experimental evidence (e.g., from IR) to verify of bonding configuration of key species such as *NO₃ and *NO? For example, does N in *NO prefer to bond to Ti or Fe?
4. In Figure 5d, the authors argue that "during the interaction between Fe and NO, electrons are inclined to be contributed to the unoccupied π* orbitals of NO. Since this process does not require overcoming the repulsion between electrons ...". However, for the case of "without spin alignment", the electron correlation and spin pairing can favor the electron coupling as well. Do you have any further computational evidence (e.g., energy different for different spin configurations) or literature data to support your argument?

Reviewer #2

(Remarks to the Author)

This is a very interesting manuscript reporting the effects of an external magnetic field on the electrocatalytic performance of Fe-doped TiO₂ for nitrate reduction to ammonia.

I think the experimental results and observations by the authors are very important. Unfortunately, I do not think the authors have analyzed properly the data, and make some assignments that are not consistent with the experiments, and some of them not even with the behavior of magnetic materials.

Therefore, my advice is for the authors to reformulate the study, and reach a solid discussion to correlate the observations with a reasonable hypothesis. These are the major concerns I have about this manuscript:

- 1) The magnetic behavior of the materials is not well determined. The appearance of hysteresis is the only data the authors claim to support "ferromagnetism". But it is not clear if the hysteresis is actually coming from the whole material. Is this a homogeneous or heterogeneous material? A RT hysteresis could appear from Fe or Fe oxide nanoparticles. This is unclear. The major part of the material is paramagnetic, not showing any magnetic ordering. Without a well-characterized material, any further hypothesis is misleading at least.

2) Then, if a material is "ferromagnetic", at least part of it, a magnetic field is only going to affect the domain boundaries. The authors statement: "The application of an external magnetic field enhances the total magnetic polarization and density of aligned magnetic moments in ferromagnetic materials," is wrong. The material is not going from antiferro to ferromagnetic. It is ferromagnetic all the time. Therefore, the DFT calculations looking at AFM and FM configurations are not related in any case to the present observations. There is no AFM state to start with. Thus the computational analysis appears to be irrelevant.

3) I am missing a very important experiment. The chronoamperometries (Fig. S38) should be performed at on/off magnetic fields. Is the effect of magnetic fields permanent? How are the dynamics as a function of time? Is there an activation time? Memory effect? These are very important questions to be answered.

On a final note, I would also ask the authors to look with detail to their own results. they state: "Notably, the 1.1 wt% Fe-TiO₂ catalyst, exhibiting solely paramagnetism, exhibited negligible magnetic effects under an applied magnetic field (Supplementary Figs. 42-44)." But if a reader looks to Fig 42-44, it is clear that the magnetic field is actually also affecting the performance of the 1.1% and of the 2.7 % samples. How can anyone think this is "negligible". It is weak, but measurable.

In summary, I think the authors are investigating an interesting system with interesting effects under magnetic fields. But neither the catalyst nor the electrochemical performance have been investigated properly, with many missing data and lacking proper analysis. Leaving all claims, and computation, without solid and consistent support.

Reviewer #3

(Remarks to the Author)

This work reports using ferromagnetic iron-embedded titanium dioxide catalysts (Fe-TiO₂) to investigate the magnetic field effects on NO₃RR.

Authors observe a ~100% increase in current density (at -0.9 V, fig 3a) under an applied magnetic field, compared to the non-magnetized situation. Additionally, they observe a matching increase in the NH₃ yield rate. The Faradaic efficiency is ~10% higher (80% no magnet, 90% magnet, -0.7 V). The authors achieved this enhancement with a ferromagnetic catalyst that has an intermediate spin state (conclusion based on incomplete coordination and magnetometry measurements.) The authors remarkably report in-situ characterization measurements in a magnetic field.

The work is original and makes a significant contribution to the electrocatalysis field. It is unique in providing magnetic field measurements during spectroscopic techniques and in studying magnetic effects on NO₃-RR.

Despite the novelty, a few significant points need attention before the manuscript can be recommended for publication.

In general, the manuscript is written as a report, and there is very little discussion and connection between the results, literature, etc. The discussion takes half a page. In such a novel and important topic, much more depth is needed.

The experimental procedures are very limited and lack important information. It would be impossible to repeat the experiments elsewhere. For the ATR, for example, there is no information on the prism, spectra acquisition details, etc. For Figure 3e, there is no clarity on what cycle means and how the experiments are performed. There is no information about the reproducibility of the results (how many repetitions were performed for each measurement, etc.).

On the FTIR, figure 4 - some of the bands are incredibly weak to be meaningful, while others, more pronounced are not explored (for example, NO, NH₄, NO₃- before NO₂-~1300, very weak; before NH ~1080 not discussed)

Supplementary Figures 28 and 38 show that the current seems to change with time. Sometimes, it increases, and sometimes, it decreases, and different behavior is observed with the potential. This is not in line with the observations in the stability tests.

Supplementary figure 34/46 - EIS data should be fitted. It looks like more than the charge transfer resistance is changing.

Moreover, all EIS data lack context and important information to ensure reproducibility: potential, frequency, and electrolyte.

Supplementary Figure 42/43/44 - However, the activity of the lower Fe loading samples under a magnetic field is surprising.

Why do those also increase, since these are not FM? The magnitude of the increase also seems to be linked to material activity, indicating mass transport enhancement might occur simultaneously. Often, more than one effect occurs under a magnetic field, but the authors failed to discuss or address these reported effects. The XAS data is remarkable and shows some differences, but it does not exclude other magnetic field effects that take place at the same time.

Version 1:

Reviewer comments:

Reviewer #1

(Remarks to the Author)

The authors have addressed all my questions and comments. The manuscript is significantly improved. I would suggest it to be accepted in the journal of Nature Communications.

Reviewer #2

(Remarks to the Author)

The authors have revised the manuscript with additional experiments, and revising the description. Even if I still have some

doubts about the actual , genuine nature of the materials, I also believe the author did their best to address all initial concerns, and the results are of great interest, enough to justify publication of this work.

Reviewer #3

(Remarks to the Author)

The authors have thoroughly revised the manuscript following the referee's comments. The ambiguous parts of the manuscript are now clear, and the results are supported by proper data and discussion. I recommend the publication of the manuscript.

Reviewer #1 (Remarks to the Author):

In this manuscript, the authors reported a new type of ferromagnetic spin electrocatalyst: Fe-embedded TiO₂ (Fe-TiO₂). They found that Fe-TiO₂ showed significantly higher NH₃ yield and Faradaic efficiency for the electrocatalytic reduction of nitrate (NO₃RR), under the influence of magnetic field effect (MFE). The authors have conducted comprehensive analysis for the Fe-TiO₂ structure and investigated the mechanism of MFE by the Fe-TiO₂. The research is a significant advancement on studying magnetic electrocatalysis for NO₃RR. I would suggest the manuscript to be accepted in the journal of Nature Communications, after a minor revision.

1. By XANES, the authors found the pattern of -Ti-O-Fe-O-Ti- in the Fe-TiO₂ catalyst. Can other atomic configurations such as -Ti-O-Fe-O-Fe- and -Ti-O-Ti-O-Fe- be present as well? Please provide evidence for that. If so, how can you control the homogeneity of structure?

Figure R1 shows three atomic configurations: -Ti-O-Fe-O-Ti-, -Ti-O-Fe-O-Fe-, and -Fe-O-Ti-O-Fe-. At lower loadings, due to the lower density, the incorporated Fe atoms are predominantly isolated and dispersed as -Ti-O-Fe-O-Ti-. As the loading increases, the distance between Fe atoms decreases. When the loading reaches 6.8 wt%, a higher doping density results in the formation of a greater number of -Fe-O-Ti-O-Fe- configurations (Figure R2). Despite the increasing doping density, the -Ti-O-Fe-O-Fe- configuration is not formed. This can be attributed to the following: For the -O-Ti-O-Ti-O- structure, when one Ti site is replaced by Fe (-O-Fe-O-Ti-O-) and the adjacent Ti is also replaced by Fe (-O-Fe-O-Fe-O-), the overall cation valence state is observed to be +6. However, if the adjacent Ti is not replaced by Fe (-O-Fe-O-Ti-O-), the cation valence state is determined to be +7, which aligns more closely with the original valence state of TiO₂. Therefore, the formation of the -Ti-O-Fe-O-Fe- structure is unlikely. Fourier transform of EXAFS spectra (Fig. 1d) at 2.5 Å failed to detect Fe-O-Fe scattering peaks, and both aberration-corrected electron microscopy and EELS analyses did not reveal the presence of adjacent Fe dimers. Additionally, the -Ti-O-Ti-O-Fe-

(single atom) configuration is unavoidable during synthesis, as confirmed by synchrotron radiation, AC-HAADF-STEM, and EELS.

Fig. R1 The top views of the Fe-TiO₂ model. (a) -Fe-O-Ti-O-Fe-, (b) -Ti-O-Fe-O-Fe-, and (c) -Ti-O-Ti-O-Fe-.

Fig. R2 Aberration-corrected HAADF-STEM image of 6.8 wt% Fe-TiO₂.

2. On Page 13, the authors mentioned “this observation suggests that within the catalysts, active sites comprising Fe species experience enhanced electron density accumulation under the influence of a magnetic field, thereby facilitating efficient electron transfer from Fe to reaction intermediates”. Could the authors comment on why electron density accumulated on Fe under magnetic field? Can you link to computational evidence to support that?

The influence of magnetic fields on electron spin and orbital motion induces alignment of electron spins, thereby altering electron distribution and orbital energy levels, which in turn affects the accumulation of electron density on Fe atoms. Quantum

spin exchange interactions (QSEI), related to the configuration of unpaired electrons, are present in both closed-shell and open-shell catalysts. QSEI play a crucial role in reshaping orbitals and catalytic structures and are significant factors affecting bond breaking and formation as well as electron transport.

In open-shell structures, characterized by more than one unfilled subshell, charge transfer during chemical reactions is influenced by electron spin. Interactions between reactants and intermediates differ markedly in ferromagnetic and antiferromagnetic structures. In ferromagnetic catalysts, the disparity in the number of spin \uparrow and spin \downarrow electrons in the outer shell increases QSEI among atoms, creating Fermi holes at the valence band of the ferromagnetic active centers. Consequently, spin delocalization exchange is accelerated, and electron repulsion is reduced, allowing electrons with the same spin to exchange orbitals more frequently. This additional stabilization energy, combined with vacant valence orbitals, enhances catalytic activity through quantum interactions. In antiferromagnetic (AFM) coupled spin configurations, no additional interatomic QSEI open shells are present, and Coulomb repulsion among atoms persists in the catalyst. Typically, the Jahn-Teller effect in antiferromagnetic materials helps reduce electron repulsion [1, 2]. However, during catalytic reactions, the Jahn-Teller effect in antiferromagnetic materials often occurs in high-energy antibonding orbitals, destabilizing the lowest unoccupied orbitals and thereby hindering electron transport. Consequently, antiferromagnetic materials exhibit inferior catalytic performance compared to ferromagnetic materials [3].

To characterize the charge differences in Fe-TiO₂ with different spin configurations, charge density difference of Fe-TiO₂, charge density difference of *NO, and Bader charge analysis have been included in the Manuscript and Supplementary Information. It is demonstrated that the impact of different spin configurations on charge distribution and transfer is significant. The specific content is as follows:

The synergy of magnetic fields is one of the primary methods reported for spin regulation⁴¹⁻⁴³. To characterize the charge differences in Fe-TiO₂ with different spin configurations, the charge density differences of Fe-TiO₂ were calculated. It is indicated

that the redistribution of d-orbital electrons in Fe atoms increases the electron density in certain orbitals, thereby enhancing their ability to transfer electrons to reaction intermediates (Supplementary Fig. 49). Additionally, the differential charge density of *NO and Bader charge analysis further demonstrate that the charge transfer between Fe and *NO can be regulated by different spin configurations (Supplementary Fig. 50 and 51).

Supplementary Fig. 49 | The charge density difference of different spin configurations Fe-TiO₂. The isosurface value is set to be 0.01 e/Bohr³.

Supplementary Fig. 50 | The charge density difference of different spin configurations NO adsorbed on Fe-TiO₂. The isosurface value is set to be 0.01 e/Bohr³.

Supplementary Fig. 51 | The charge transfer of different spin configurations NO adsorbed on Fe-TiO₂.

Reference:

1. Halcrow, M. A. Jahn–teller distortions in transition metal compounds, and their importance in functional molecular and inorganic materials. *Chem. Soc. Rev.* **42**, 1784-

1795 (2013).

2. Biz, C. et al. Strongly Correlated electrons in catalysis: focus on quantum exchange. *ACS Catal.* **11**, 14249–14261 (2021).

3. Bersuker, I. B. Jahn–teller and pseudo-jahn–teller effects: from particular features to general tools in exploring molecular and solid state properties. *Chem. Rev.* **121**, 1463–1512 (2021).

41. Huang, J. et al. Enhancing hydrogen evolution reaction of confined monodispersed NiSe_{2-x} nanoparticles by high-frequency alternating magnetic fields. *Chem. Eng. J.* **454**, 140279(2023).

42. Li, Y. et al. Electrocatalytic reduction of low-concentration nitric oxide into ammonia over Ru nanosheets. *ACS Energy Lett.* **7**, 1187-1194 (2022).

43. Zhang, Y. et al. Recent advances in magnetic field-enhanced electrocatalysis. *ACS Appl. Energy Mater.* **3**, 10303-10316 (2020).

3. Figure 5 shows the bonding configurations of reactant/product and intermediate species on Fe-TiO₂ surface. Do you have experimental evidence (e.g., from IR) to verify of bonding configuration of key species such as *NO₃ and *NO? For example, does N in *NO prefer to bond to Ti or Fe?

To experimentally confirm the adsorption of nitrogen atoms on Fe, X-ray emission spectroscopy (XES) was conducted on the catalyst before and after the reaction. At higher emission energies, valence electron transitions to the metal 1s core hole (i.e., the Kβ_{2,5}/Kβ'' or V2C region) can be observed. These transitions are attributed to ligand np → metal 1s (Kβ_{2,5}, ~7110 eV) and ligand ns → metal 1s (Kβ'', ~7095 eV) transitions. Therefore, Kβ'' is typically used to identify information about different ligands bound to the metal, particularly distinguishing elements such as nitrogen (N), oxygen (O), and carbon (C). As shown in the Supplementary Fig. 57, after 1 hour of reaction, a significant shift in the Kβ'' position (from 7092.7 eV to 7094.7 eV) was observed. Based on previous studies [21], this shift is attributed to the adsorption of nitrogen atoms on Fe. Therefore, it is concluded that Fe serves as the adsorption site for N.

Supplementary Fig. 57 | The X-ray emission spectroscopy of Fe-TiO₂.

Reference:

1. Lancaster K. M. et al. X-ray emission spectroscopy evidences a central carbon in the nitrogenase iron-molybdenum cofactor. *Science* **334**, 974-977 (2011).

4. In Figure 5d, the authors argue that “during the interaction between Fe and NO, electrons are inclined to be contributed to the unoccupied π^* orbitals of NO. Since this process does not require overcoming the repulsion between electrons ...”. However, for the case of “without spin alignment”, the electron correlation and spin pairing can favor the electron coupling as well. Do you have any further computational evidence (e.g., energy different for different spin configurations) or literature data to support your argument?

Different spin configurations, including spin states: $S = 0, 2, 4, 6, 8,$ and 10 , were selected for the calculation of surface energies and complete reaction pathways. It was found that surface energies are significantly affected by spin configurations, with the SPIN=8 configuration being the most stable. Additionally, substantial impacts on the energy differences of reaction intermediates were observed due to spin configurations. Notably, during *NO adsorption, the lowest reaction barrier was exhibited by the high spin states ($S = 8$) configuration, identifying it as the optimal spin configuration.

The following discussions were included in the revised manuscript (page. 16):

To elucidate the origin of the magnetic field effect (MFE) in Fe-TiO₂, various spin

configurations of Fe-TiO₂, including spin states $S = 0, 2, 4, 6, 8,$ and $10,$ were investigated⁴⁸⁻⁵⁰. These configurations, representing distinct magnetic states, were analyzed to facilitate an understanding of the role of MFE in the catalytic performance of Fe-TiO₂. Furthermore, the influence of spin configurations on material properties was systematically examined by calculating surface energies, reaction pathways, and energy changes. It was indicated that spin configurations significantly affect surface energy, with the high spin states ($S=8$) configuration being the most stable (Supplementary Fig. 55). Additionally, the entire reaction pathways for different spin configurations were calculated. Significant energy differences for various intermediates were observed, with the high spin states ($S=8$) configuration exhibiting the lowest reaction barrier, identifying it as the optimal spin configuration (Supplementary Fig.56). Based on this, Gibbs free energy calculations for two representative magnetic structures, Fe-TiO₂ with spin alignment and Fe-TiO₂ without spin alignment, were performed, as shown in Fig. 5a⁵¹.

Supplementary Fig. 55 | Surface energy for different spin configurations.

Supplementary Fig. 56 | Free energy diagram for nitrite electroreduction on Fe-TiO₂ under different spin configurations.

Reference:

45. Zhong, W. et al. Electronic spin moment as a catalytic descriptor for Fe single-atom catalysts supported on C₂N. *J. Am. Chem. Soc.* **143**, 4405-4413 (2021).
46. Zhang, Y. et al. Spin states of metal centers in electrocatalysis. *Chem. Soc. Rev.* **53**, 8123-8136 (2024).
47. Luo, S. et al. Electrochemistry in magnetic fields. *Angew. Chem. Int. Edit.* **61**, e202203564 (2022).
48. Ren, X. et al. Spin-polarized oxygen evolution reaction under magnetic field. *Nat. Commun.* **12**, 2608 (2021).

Reviewer #2 (Remarks to the Author):

This is a very interesting manuscript reporting the effects of an external magnetic field on the electrocatalytic performance of Fe-doped TiO₂ for nitrate reduction to ammonia. I think the experimental results and observations by the authors are very important. Unfortunately, I do not think the authors have analyzed properly the data, and make some assignments that are not consistent with the experiments, and some of them not even with the behavior of magnetic materials. Therefore, my advice is for the authors to reformulate the study, and reach a solid discussion to correlate the observations with a reasonable hypothesis. These are the major concerns I have about this manuscript:

1) The magnetic behavior of the materials is not well determined. The appearance of hysteresis is the only data the authors claim to support "ferromagnetism". But it is not clear if the hysteresis is actually coming from the whole material. Is this a homogeneous or heterogeneous material? A RT hysteresis could appear from Fe or Fe oxide nanoparticles. This is unclear. The major part of the material is paramagnetic, not showing any magnetic ordering. Without a well-characterized material, any further hypothesis is misleading at least.

In addition to the hysteresis loops, the M-T curves of the catalyst materials are presented. It is demonstrated that 6.8 wt% Fe-TiO₂ exhibits ferromagnetism. As shown in Fig. R3, the M-T curve for 1.1 wt% Fe-TiO₂ shows a χ value approaching infinity near 0K, characteristic of paramagnetic materials. In contrast, the χ value for 6.8 wt% Fe-TiO₂ approaches saturation, near 0K, further confirming its ferromagnetic nature [1].

Fig. R3 The M-T curves of the catalyst materials. (a) 1.1 wt% Fe-TiO₂ and (b). 6.8 wt% Fe-TiO₂.

Simultaneously, the material was found to be homogeneous, containing only single-atom sites, as no presence of Fe or Fe oxide nanoparticles was revealed by our characterization techniques. In XPS (Fig. R4a), no characteristic peak at 707 eV corresponding to Fe metal was observed [2], thus preliminarily excluding the presence of Fe metal. HAADF STEM EDS mapping (Fig. R4b) also did not show any aggregation of Fe metal or iron oxide particles. Furthermore, high-resolution AC-STEM (Fig. R4c) did not detect small Fe metal or iron oxide clusters. To rule out the possibility of undetected Fe metal or iron oxide particles due to limitations of aberration-corrected electron microscopy, we conducted further analysis using EXAFS on the bulk phase. The fitting of the R-space data (Fig. R4d) indicated the absence of Fe-Fe bonds (2.2 and 2.5 Å) in the bulk phase of our catalyst.

Furthermore, no reduction processes were involved in the synthesis method, and extensive acid washing with concentrated hydrochloric acid was employed during protonation (1 M HCl, 3 times). Consequently, the presence of elemental Fe and iron oxide particles should be precluded. Therefore, the magnetic properties of the catalyst are not attributed to Fe metal or iron oxide particles.

Fig. R4 (a) and Fe 2p XPS High-Resolution Spectrum of Fe-TiO₂. (b) EDS elemental mapping of Fe-TiO₂ catalysts. (c) Aberration-corrected HAADF-STEM image, where a single Fe atom is circled in red color. The inset on the left displays the energy loss spectrum of the selected dashed square. The inset above shows the intensity profiles. (d) Fourier transform of EXAFS spectra of Fe foil, Fe₂O₃ and Fe-TiO₂.

As shown in Fig. 2b, a combination of paramagnetic and ferromagnetic properties is exhibited by 6.8 wt% Fe-TiO₂. The paramagnetism is provided by isolated Fe single atoms, while the ferromagnetism arises from closely spaced Fe single atoms at high doping densities. The effect of an external magnetic field on nitrate reduction under ferromagnetic conditions is the focus of this research. To better understand the contribution of ferromagnetism to the magnetic field effect (MFE) and to distinguish between the contributions of paramagnetism and ferromagnetism to MFE, materials with paramagnetism and weak ferromagnetism (2.7 wt% Fe-TiO₂) as well as purely paramagnetic materials (1.1 wt% Fe-TiO₂) were synthesized, and corresponding nitrate reduction performance tests under a magnetic field were conducted. As shown in Figs.

S43-45, a diminished MFE is observed with weaker ferromagnetism, and when ferromagnetism is completely absent, leaving only paramagnetism, the MFE nearly vanishes. Therefore, it is demonstrated by comparative experiments that only ferromagnetism results in a significant magnetic field effect. In other words, the predominant contribution to the magnetic effect is attributed to the ferromagnetic component of the material, which is the primary focus of this study.

Supplementary Fig. 42 | The LSV curves of Fe-TiO₂ with different loading were obtained with and without the magnetic field opened. The solid line and the dotted line represent with and without the magnetic field opened., respectively.

Supplementary Fig. 43 | Diagram of electrochemical testing under a magnetic field. NH₃ yield (a) and FE_{NH₃} (b) of Fe-TiO₂ (1.1 wt%) catalyst at various potentials in the presence or absence of 300 mT external magnetic fields.

Supplementary Fig. 44 | Diagram of electrochemical testing under a magnetic field. NH₃ yield (**a**) and FE_{NH₃} (**b**) of Fe-TiO₂ (2.7 wt%) catalyst at various potentials in the presence or absence of 300 mT external magnetic fields.

Reference:

1. Sarma S. D. et al. Temperature-dependent magnetization in diluted magnetic semiconductors. *Phys. Rev. B* **67**, 286-293 (2003).
2. Luna M. L. et al. Role of the Oxide Support on the Structural and Chemical Evolution of Fe Catalysts during the Hydrogenation of CO₂. *ACS Catal.* **11**, 6175-6185 (2021).

2) Then, if a material is "ferromagnetic", at least part of it, a magnetic field is only going to affect the domain boundaries. The authors statement: "The application of an external magnetic field enhances the total magnetic polarization and density of aligned magnetic moments in ferromagnetic materials," is wrong. The material is not going from antiferro to ferromagnetic. It is ferromagnetic all the time. Therefore, the DFT calculations looking at AFM and FM configurations are not related in any case to the present observations. There is no AFM state to start with. Thus the computational analysis appears to be irrelevant.

An error was identified in our initial submission. The statement, "The application of an external magnetic field enhances the total magnetic polarization and density of aligned magnetic moments in ferromagnetic materials," has been deleted. Additionally, incorrect descriptions of AFM and FM were provided. The aim of the theoretical

calculations was to elucidate the impact of the magnetic field on the overall reaction, specifically identifying which step is promoted by spin polarization under the magnetic field. In light of the reviewer's comments and our understanding of existing research, our descriptions have been revised: AFM should be represented as without spin alignment and FM as with spin alignment [1]. The entire manuscript has been revised to ensure accuracy in these descriptions. Furthermore, a discussion on the NO₃RR reaction pathways under different spin configurations has been added (see Reviewer #1, point 4).

Reference:

1. Ren, X. et al. Spin-polarized oxygen evolution reaction under magnetic field. *Nat. Commun.* **12**, 2608 (2021).

3) I am missing a very important experiment. The chronoamperometries (Fig. S38) should be performed at on/off magnetic fields. Is the effect of magnetic fields permanent? How are the dynamics as a function of time? Is there an activation time? Memory effect? These are very important questions to be answered.

Based on the reviewer's comments, the following experimental data and analysis have been supplemented: As shown in Fig. R5, when the magnetic field is turned on or off, the change in current density is initiated immediately, but tens of seconds are required to reach a new steady state. The time constants for the exponential rise and fall were calculated. For 1.1 wt% Fe-TiO₂ (paramagnetic catalyst), the rise and fall time constants of the MFE transient dynamics are nearly identical, approximately 13 s. In contrast, for the 6.8 wt% Fe-TiO₂ electrode (ferromagnetic catalyst), the rise and fall time constants of the MFE are approximately 54.2 s and 40.9 s, respectively. This significant discrepancy suggests that an inherent "hysteresis" effect is present in 6.8 wt% Fe-TiO₂ between the turning on and off of the magnetic field. Additionally, when the external magnetic field is reduced to zero, a relaxation time is required for the system to return to its original state, which is attributed to the inherent hysteresis of the ferromagnetic catalyst. Therefore, it is demonstrated that the magnetic field effect is not permanent, with an activation time exceeding 40 s.

Fig. R5 The i - t curves (at -0.3 V vs. Ag/AgCl) showing changes in the current density under magnetic fields (300 mT) using (a) 1.1 wt% Fe-TiO₂ and (b) 6.8 wt% Fe-TiO₂ catalysts in the electrochemical reactions.

4) On a final note, I would also ask the authors to look with detail to their own results. they state: "Notably, the 1.1 wt% Fe-TiO₂ catalyst, exhibiting solely paramagnetism, exhibited negligible magnetic effects under an applied magnetic field (Supplementary Figs. 42-44)." But if a reader looks to Fig 42-44, it is clear that the magnetic field is actually also affecting the performance of the 1.1% and of the 2.7 % samples. How can anyone think this is "negligible". It is weak, but measurable.

An error is present in our statement within the manuscript. The NH₃ yield and Faradaic efficiency of 1.1%, 2.7%, and 6.8% Fe-TiO₂, with and without the magnetic field, are summarized in Supplementary Table 3-5, detailing the observed changes. It is indicated by the results that the magnetic field has a minimal impact on the NH₃ yield and Faradaic efficiency of 1.1 wt% and 2.7 wt% Fe-TiO₂, which is significantly less than its impact on 6.8% Fe-TiO₂.

Under low Fe loading conditions (i.e., paramagnetic materials), there indeed exists a non-negligible magnetic field effect, as pointed out by the reviewer (Reviewer #3, point 6). When studying the magnetic field effect in electrocatalysis, the optimization of the mass transfer process under the influence of the Lorentz force cannot be ignored. Although we have tried to minimize this effect (by orienting the electrode surface perpendicular to the magnetic field lines so that the movement direction of charged

particles is parallel to the magnetic field lines and not affected by the Lorentz force), the electrode surface is not perfectly smooth and still contains some rough "trenches". Therefore, the influence of the Lorentz force, though small, still exists.

To address this, we conducted comparative experiments. The results show that when using 6.8 wt% catalyst and aligning the electrode surface parallel to the magnetic field lines, the performance enhancement further increased (although the increase was very slight compared to the enhancement observed from 0 to 300 mT). When testing with 1.1 wt% catalyst, the performance increase also further improved when the electrode surface was parallel to the magnetic field lines. Hence, it can be concluded that part of the performance enhancement observed in Supplementary Fig. 45 is attributable to the influence of the Lorentz force.

Although the influence of the Lorentz force contributes to a certain degree of performance enhancement, its effect is far smaller than the significant enhancement observed in ferromagnetic materials, indicating that the two effects are not of the same magnitude.

The following discussions were included in the revised Supplementary information:

The minimal impact on NH₃ yield and Faradaic efficiency under an external magnetic field is exhibited by the 1.1 wt% Fe-TiO₂ catalyst, which shows only paramagnetism, significantly less than the effect observed for 6.8 wt% Fe-TiO₂ (Supplementary Figs. 42-44, Tables 3-5). This may be attributed to the optimization of mass transfer processes under the influence of Lorentz forces; however, the effect is considerably smaller than the significant enhancement observed in ferromagnetic materials (Supplementary Fig. 45, Tables 6, 7).

Supplementary Table 3. Catalytic performance of 1.1wt% Fe-TiO₂.

NH ₃ yield rate (mg mg _{cat} ⁻¹ h ⁻¹)	Faraday efficiency (%)
--	------------------------

	Without	With	growth rate	Without	With	growth rate
-0.3 V	0.9	1.0	5.6%	57.0	58.0	1.8%
-0.4 V	1.3	1.4	7.7%	58.0	59.0	1.7%
-0.5 V	2.6	2.7	3.8%	65.0	66.0	1.5%
-0.6 V	3.8	3.9	3.4%	65.0	67.0	3.1%
-0.7 V	5.0	5.4	7.0%	66.0	67.0	1.5%

Supplementary Table 4. Catalytic performance of 2.7wt% Fe-TiO₂.

	NH ₃ yield rate (mg mg _{cat} ⁻¹ h ⁻¹)			Faraday efficiency (%)		
	Without	With	growth rate	Without	With	growth rate
-0.3 V	1.2	1.5	25.0%	68.0	71.0	4.4%
-0.4 V	2.0	2.5	25.0%	70.0	73.0	4.3%
-0.5 V	4.1	5.5	34.1%	73.0	75.5	3.4%
-0.6 V	6.3	8.3	31.7%	73.0	76.5	4.8%
-0.7 V	8.0	11.5	43.8%	74.0	77.0	4.1%

Supplementary Table 5. Catalytic performance of 6.8wt% Fe-TiO₂.

	NH ₃ yield rate (mg mg _{cat} ⁻¹ h ⁻¹)			Faraday efficiency (%)		
	Without	With	growth rate	Without	With	growth rate
-0.3 V	1.7	4.2	150.0%	71.7	81.4	13.5%
-0.4 V	3.0	11.9	300.0%	72.3	90.5	25.2%

-0.5 V	5.9	24.6	314.3%	80.1	97.6	21.9%
-0.6 V	10.2	30.2	195.6%	82.7	90.0	8.9%
-0.7 V	16.2	34.8	114.7%	83.3	90.4	8.5%

Supplementary Fig. 45| Electrochemical performance under varying magnetic field orientations. Dashed lines represent data in the without magnetic field: (a) 1.1 wt%, (b) 6.8 wt% Fe-TiO₂.

Supplementary Table 6. Catalytic performance under varying magnetic field orientations of 1.1 wt% Fe-TiO₂.

	NH ₃ yield rate (mg mg _{cat} ⁻¹ h ⁻¹)			Faraday efficiency (%)		
	Without	With	growth rate	Without	With	growth rate
0	2.6	2.8	7.7%	65	66.3	2.0%
90	2.6	2.7	3.8%	65	65.2	0.3%
180	2.6	2.77	6.5%	65	66.1	1.7%

Supplementary Table 7. Catalytic performance under varying magnetic field orientations of 6.8 wt% Fe-TiO₂.

	NH ₃ yield rate (mg mg _{cat} ⁻¹ h ⁻¹)			Faraday efficiency (%)		
	Without	With	growth rate	Without	With	growth rate
0	5.9	25.4	330.5%	80	98	22.5%
90	5.9	24.6	316.9%	80	97.6	22.0%
180	5.9	25.5	332.2%	80	98	22.5%

In summary, I think the authors are investigating an interesting system with interesting effects under magnetic fields. But neither the catalyst nor the electrochemical performance have been investigated properly, with many missing data and lacking proper analysis. Leaving all claims, and computaion, without solid and consistent support.

Reviewer #3 (Remarks to the Author):

This work reports using ferromagnetic iron-embedded titanium dioxide catalysts (Fe-TiO₂) to investigate the magnetic field effects on NO₃RR. Authors observe a ~100% increase in current density (at -0.9 V, fig 3a) under an applied magnetic field, compared to the non-magnetized situation. Additionally, they observe a matching increase in the NH₃ yield rate. The Faradaic efficiency is ~10% higher (80% no magnet, 90% magnet, -0.7 V). The authors achieved this enhancement with a ferromagnetic catalyst that has an intermediate spin state (conclusion based on incomplete coordination and magnetometry measurements.) The authors remarkably report in-situ characterization measurements in a magnetic field. The work is original and makes a significant contribution to the electrocatalysis field. It is unique in providing magnetic field measurements during spectroscopic techniques and in studying magnetic effects on NO₃RR. Despite the novelty, a few significant points need attention before the manuscript can be recommended for publication.

1) In general, the manuscript is written as a report, and there is very little discussion and connection between the results, literature, etc. The discussion takes half a page. In such a novel and important topic, much more depth is needed.

We have updated the full manuscript to increase the depth of the manuscript.

2) The experimental procedures are very limited and lack important information. It would be impossible to repeat the experiments elsewhere. For the ATR, for example, there is no information on the prism, spectra acquisition details, etc. For Figure 3e, there is no clarity on what cycle means and how the experiments are performed. There is no information about the reproducibility of the results (how many repetitions were performed for each measurement, etc.).

Experimental details on ATR and stability cycling tests have been added to the revised manuscript (page 21 and 22). The specific content is as follows:

During the collection of ATR-FTIR spectra, an attenuated total reflectance (ATR) prism made of wedge-shaped Si crystal was utilized, with an incident angle of 60 degrees. All measurements were obtained through 64 scans with a spectral resolution of 4 cm^{-1} . Chronoamperometry tests at OCP to -0.5 V RHE were conducted on a CHI 660E electrochemical workstation, and spectra were collected.

For the stability cycling tests, the catalyst loading, coating area, electrolyte composition, and volume were kept identical to previous conditions. The test potential was set to 0.5 V , with an external magnetic field of 300 mT , using the same H-type electrolytic cell as before. Each test lasted for 1 hour, after which the electrolyte was collected for colorimetric analysis and calculation of yield and Faradaic efficiency. Three samples were taken and analyzed, with the results averaged. This constituted one cycle. After each cycle, the electrode was gently rinsed with fresh electrolyte, and the next cycle was conducted with new electrolyte.

3) On the FTIR, figure 4 - some of the bands are incredibly weak to be meaningful, while others, more pronounced are not explored (for example, NO , NH_4 , NO_3^- before NO_2^- ~1300, very weak; before NH ~1080 not discussed)

To more intuitively understand and distinguish the effects of the magnetic field on the electrocatalytic nitrate reduction reaction for ammonia synthesis, Fig. 4b and 4c have been updated, and Supplementary Fig. 49 has been added, with particular attention given to the characteristic peaks before NH ~1080 cm^{-1} . The corresponding part of the manuscript has been updated (page. 15 and 16). The specific testing details are as follows:

The synergy of magnetic fields is one of the primary methods reported for spin regulation⁴¹⁻⁴³. To characterize the charge differences in Fe-TiO_2 with different spin configurations, the charge density differences of Fe-TiO_2 were calculated. It is indicated that the redistribution of d-orbital electrons in Fe atoms increases the electron density in certain orbitals, thereby enhancing their ability to transfer electrons to reaction

intermediates (Supplementary Fig. 49). Additionally, the differential charge density of *NO and Bader charge analysis further demonstrate that the charge transfer between Fe and *NO can be regulated by different spin configurations (Supplementary Fig. 50 and 51).

To further investigate the influence of the magnetic field on the entire reaction process, *in situ* ATR-FTIR spectroscopy was employed to detect intermediates with and without an applied magnetic field. Initially, experiments were performed using TiO₂ and Fe-TiO₂ in 0.1 M KOH and 0.1 M KNO₃ electrolytes, covering a potential range from open circuit potential (OCP) to -0.5 V vs. RHE. For Fe-TiO₂ (Fig. 4d and Supplementary Fig. 52a), as the potential became more negative, a series of characteristic peaks corresponding to reaction intermediates appeared in the ATR-FTIR spectra: N–O stretching vibration of NO₂⁻ ions (1227 cm⁻¹)¹⁵, NOH species (1156 cm⁻¹)⁴⁴, and NH species (1050 cm⁻¹)⁴⁵. These peaks intensified progressively, indicating an enhancement in the reaction, consistent with electrochemical testing results. Concurrently, a series of weaker characteristic peaks were observed, including a negative peak at 1380 cm⁻¹ (corresponding to the N–O stretching vibration of adsorbed NO₃⁻ ions)⁴⁶, the N–O bending vibration in NO at 1567 cm⁻¹⁴⁷, and the N–H bending vibration of NH₄⁺ ions at 1467 cm⁻¹ (Supplementary Fig. 53b)⁴⁶. The gradual enhancement of these characteristic peaks indicates an intensifying reaction that is consistent with electrochemical testing observations and aligned with theoretical reaction steps. In contrast, for pure TiO₂ (Supplementary Fig. 54), weak characteristic peaks related to NO₃⁻ were observed through ATR-FTIR spectroscopy, implying inadequate adsorption capacity for pure TiO₂ towards NO₃⁻, thereby hindering completion of the reaction pathway (the presence of very weak characteristic peaks related to NH₄⁺ and other intermediates further supports this). These results demonstrate that Fe serves as the exclusive active site for NO₃RR within Fe-TiO₂.

Subsequently, Fe-TiO₂ was further investigated to elucidate the underlying mechanism of the observed MFE (Fig. 4c and Supplementary Fig. 52b, 53a). Upon application of magnetic field, significant changes in the characteristic peaks of reaction

intermediates were observed. The NO peak disappeared, the intensity of the *NOH species vibration at 1156 cm⁻¹ was notably increased, and the *NH species vibration at 1050 cm⁻¹ was also intensified. Additionally, a distinct *NH₂ peak appeared at 1439 cm⁻¹. These changes indicate that the reaction proceeded more rapidly under the influence of the magnetic field. It can thus be inferred that the magnetic field accelerated the hydrogenation process, converting *NO to *NOH, thereby expediting the overall reaction.

Fig. 4 | Mechanistic studies for the NO₃RR over Fe-TiO₂. Operando Fe K-edge XANES spectra for Fe-TiO₂ under 100 mT (a) and 0 mT (b). *In-situ* ATR-FTIR spectra of Fe-TiO₂ with negative scan from OCP to -0.5 V vs. RHE in the presence (c) or absence (d) of external magnetic fields.

Supplementary Fig. 49 | The charge density difference of different spin configurations Fe-TiO₂. The isosurface value is set to be 0.01 e/Bohr³.

Supplementary Fig. 50 | The charge density difference of different spin configurations NO adsorbed on Fe-TiO₂. The isosurface value is set to be 0.01 e/Bohr³.

Supplementary Fig. 51 | The charge transfer of different spin configurations NO adsorbed on Fe-TiO₂.

Supplementary Fig. 52 | The 3D *in situ* ATR-FTIR spectra of Fe-TiO₂ with negative scan from OCP to -0.5 V vs. RHE in the absence (a) or presence (b) of external magnetic fields.

Supplementary Fig. 53 | *In-situ* ATR-FTIR spectra of Fe-TiO₂ with negative scan from OCP to -0.5 V vs. RHE in the presence (c) or absence (d) of external magnetic fields.

Supplementary Fig. 54 | *In situ* ATR-FTIR Spectra of TiO₂ during Negative Scan from OCP to -0.5 V vs. RHE.

4) Supplementary Figures 28 and 38 show that the current seems to change with time. Sometimes, it increases, and sometimes, it decreases, and different behavior is observed with the potential. This is not in line with the observations in the stability tests.

We are very grateful to the reviewers for pointing out this issue and have given it thorough consideration and experimental verification.

Regarding the decrease in current, the limitation arises from the inability to stir the electrolyte due to the application of an external magnetic field. This significantly impacts the mass transfer process. At the beginning of the electrolysis, the reactant (nitrate) near the electrode is rapidly consumed at a rate greater than its replenishment, leading to a decrease in current. Over time, the mass transfer process reaches a steady state, and the current stabilizes.

Regarding the increase in current, we found that it was due to occasionally overlooking the electrode activation process. During the tests, hydrophobic carbon

paper was used, and the electrodes were inserted in a dry state, resulting in a gas-solid-liquid interface between the electrode and the electrolyte. As electrolysis progressed, the electrode surface became gradually wetted, eliminating this interface and leading to improved contact between the electrode and the electrolyte, which resulted in an increase in current. To address this, we retested under conditions where the electrode was fully wetted (Fig. R6). The results showed that, with the electrode fully wetted, the phenomenon of increasing current did not reoccur.

Fig. R6 I-t curve for Fe-TiO₂ at different potentials. (a) TiO₂, (b) Fe-TiO₂ and (c) Fe-TiO₂ with magnetic field.

5) Supplementary figure 34/46 - EIS data should be fitted. It looks like more than the charge transfer resistance is changing. Moreover, all EIS data lack context and important information to ensure reproducibility: potential, frequency, and electrolyte.

Supplementary Figures 34 and 46 (47) have been updated, and the fitting of the EIS data has been completed. The specific testing details are as follows:

EIS measurements were conducted at open circuit potential with an applied AC voltage of 5 mV, over a frequency range of 0.5 Hz to 10 kHz. The electrolyte consisted of 0.1 M KOH and 0.1 M KNO₃.

The fitting analysis of Nyquist plots reveals that the charge transfer resistance (R_{ct}) is reduced from 33.02 Ω to 7.56 Ω by Fe doping, indirectly suggesting that Fe serves as an active site for NO₃RR. Furthermore, the fitting of Nyquist plots obtained under an external magnetic field shows a further reduction of R_{ct} from 7.56 Ω to 2.66 Ω, indicating that the electron transfer rate in the reaction is significantly enhanced by the

magnetic field.

Supplementary Fig. 34 Electrochemical impedance spectroscopy (EIS) of Fe-TiO₂ and TiO₂.

Supplementary Fig. 47 Electrochemical impedance spectroscopy (EIS) of Fe-TiO₂ under no magnetic field (labeled as Fe-TiO₂) and with an open magnetic field (labeled as Fe-TiO₂-M).

Supplementary Table 8. Electrochemical impedance spectroscopy (EIS) fitting data.

	R_s (Ω)	R_{ct} (Ω)
TiO ₂	6.712	33.02
Fe-TiO ₂	6.074	7.557

Fe-TiO₂-M

5.855

2.664

6) Supplementary Figure 42/43/44 - However, the activity of the lower Fe loading samples under a magnetic field is surprising. Why do those also increase, since these are not FM? The magnitude of the increase also seems to be linked to material activity, indicating mass transport enhancement might occur simultaneously. Often, more than one effect occurs under a magnetic field, but the authors failed to discuss or address these reported effects. The XAS data is remarkable and shows some differences, but it does not exclude other magnetic field effects that take place at the same time.

Under low Fe loading conditions (i.e., paramagnetic materials), there indeed exists a non-negligible magnetic field effect, as pointed out by the reviewer. When studying the magnetic field effect in electrocatalysis, the optimization of the mass transfer process under the influence of the Lorentz force cannot be ignored. Although we have tried to minimize this effect (by orienting the electrode surface perpendicular to the magnetic field lines so that the movement direction of charged particles is parallel to the magnetic field lines and not affected by the Lorentz force), the electrode surface is not perfectly smooth and still contains some rough "trenches." Therefore, the influence of the Lorentz force, though small, still exists.

To address this, we conducted comparative experiments. The results show that when using 6.8 wt% catalyst and aligning the electrode surface parallel to the magnetic field lines, the performance enhancement further increased (although the increase was very slight compared to the enhancement observed from 0 to 300 mT). When testing with 1.1 wt% catalyst, the performance increase also further improved when the electrode surface was parallel to the magnetic field lines. Hence, it can be concluded that part of the performance enhancement observed in Supplementary Fig. 45 is attributable to the influence of the Lorentz force.

Although the influence of the Lorentz force contributes to a certain degree of performance enhancement, its effect is far smaller than the significant enhancement

observed in ferromagnetic materials, indicating that the two effects are not of the same magnitude.

The following discussions were included in the revised Supplementary information:

This may be attributed to the optimization of mass transfer processes under the influence of Lorentz forces; however, the effect is considerably smaller than the significant enhancement observed in ferromagnetic materials (Supplementary Fig. 45, Tables 6, 7).

Supplementary Fig. 45| Electrochemical performance under varying magnetic field orientations. Dashed lines represent data in the without magnetic field: (a) 1.1 wt%, (b) 6.8 wt% Fe-TiO₂.

Supplementary Table 6. Catalytic performance under varying magnetic field orientations of 1.1 wt% Fe-TiO₂.

	NH ₃ yield rate (mg mg _{cat} ⁻¹ h ⁻¹)			Faraday efficiency (%)		
	Without	With	growth rate	Without	With	growth rate
0	2.6	2.8	7.7%	65	66.3	2.0%
90	2.6	2.7	3.8%	65	65.2	0.3%
180	2.6	2.77	6.5%	65	66.1	1.7%

Supplementary Table 7. Catalytic performance under varying magnetic field

orientations of 6.8 wt% Fe-TiO₂.

	NH ₃ yield rate (mg mg _{cat} ⁻¹ h ^{-1s})			Faraday efficiency (%)		
	Without	With	growth rate	Without	With	growth rate
0	5.9	25.4	330.5%	80	98	22.5%
90	5.9	24.6	316.9%	80	97.6	22.0%
180	5.9	25.5	332.2%	80	98	22.5%